# Prevalence and determinants of severity levels of anemia among children aged 6–59 months in sub-Saharan Africa: A multilevel ordinal logistic regression analysis

**Getayeneh Antehunegn Tesema** [1]*, **Misganaw Gebrie Worku**[2], **Zemenu Tadesse Tessema**[1], **Achamyeleh Birhanu Teshale**[1], **Adugnaw Zeleke Alem**[1], **Yigizie Yeshaw** [1,3], **Tesfa Sewunet Alamneh**[1], **Alemneh Mekuriaw Liyew**[1]

1 Department of Epidemiology and Biostatistics, Institute of Public Health, College of Medicine and Health Sciences, University of Gondar, Gondar, Ethiopia, 2 Department of Human Anatomy, College of Medicine and Health Science, School of Medicine, University of Gondar, Gondar, Ethiopia, 3 Department of Physiology, School of Medicine, College of Medicine and Health Sciences, University of Gondar, Gondar, Ethiopia

* getayenehantehunegn@gmail.com

**Data Availability Statement:** All the data files are available from the measure. DHS program Data is available online and you can access it from www.

## Abstract

### Background

Anemia is a major public health problem affecting more than half of children under the age of five globally. It has serious short- and long-term consequences including growth retardation, impaired motor and cognitive development, and increased morbidity and mortality. Despite anemia is the leading cause of child mortality in sub-Saharan Africa, there is limited evidence on the prevalence and determinants of anemia among under-five children in sub-Saharan Africa. Therefore, this study aimed to investigate the prevalence and determinants of severity levels of anemia among children aged 6–59 months in sub-Saharan Africa.

### Methods

This study was based on the most recent Demographic and Health Survey (DHS) data of 32 sub-Saharan African countries. A total weighted sample of 135,619 children aged 6–59 months was included in the study. Considering the hierarchical nature of DHS data and the ordinal nature of anemia, a multilevel ordinal logistic regression model was applied. Proportional odds assumption was tested by Brant test and it was satisfied (p-value = 0.091). Besides, deviance was used for model comparison. Variables with a p-value ≤0.2 in the bivariable analysis were considered for the multivariable analysis. In the multivariable multilevel proportional odds model, the Adjusted Odds Ratio (AOR) with 95% Confidence Interval (CI) were reported for potential determinant factors of severity levels of anemia.

### Results

The overall prevalence of anemia among children aged 6–59 months in sub-Saharan Africa was 64.1% [95% CI: 63.9%, 64.4%]. Of which, 26.2% were mildly anemic, 34.9% moderately anemic and 3% severely anemic. Poor maternal education, lower household wealth

measuredhs.com. We used the Kids Record (KR) file and extract the variables based on literature. Then, we kept the same variables in all the 32 SSA countries and appended together.

**Funding:** The authors received no specific funding for this work.

**Competing interests:** The authors have declared that no competing interests exist.

**Abbreviations:** AOR, Adjusted Odds Ratio; DHS, Demographic health survey; CI, Confidence Interval; EAs, Enumeration areas; ICC, Intra-cluster Correlation Coefficient; HAZ, Z-score for Height-for-Age; IUGR, Intra uterine growth restriction; LLR, Log likelihood ratio; LR, Likelihood ratio; SDG, Sustainable Development Goal; SSA, sub-Saharan Africa; WAZ, Z-score for Weight-for-Age; WHZ, Z-score for Weight-for-Height.

status, large family size, being male child, multiple births, having fever in the last two weeks, having diarrhea in the last two weeks, higher-order birth, maternal anemia, underweight, wasted, and stunted were significantly associated with increased odds of higher levels of anemia. Whereas, being 24–59 months age, taking drugs for an intestinal parasite, and born from mothers aged $\geq$ 20 years were significantly associated with lower odds of higher levels of anemia.

## Conclusion

Severity levels of anemia among children aged 6–59 months in sub-Saharan Africa was a major public health problem. Enhancing maternal education, providing drugs for an intestinal parasite, designing interventions that address maternal anemia, febrile illness, and diarrheal disease, and strengthening the economic status of the family are recommended to reduce childhood anemia. Furthermore, it is better to strengthen the strategies of early detection and management of stunted, wasted, and underweight children to decrease childhood anemia.

## Background

Anemia is a condition characterized by an insufficient number of healthy red blood cells, often in conjunction with decreased haemoglobin levels or irregular blood cell morphology, that impairs blood from delivering oxygen to the body's tissues [1, 2]. The World Health Organization (WHO) defines anemia among under-five children as a Hb concentration below 110 g/L [3]. The childhood period is a critical period for children's growth and development, and they are more vulnerable to anemia [4]. Nutritional deficiencies (such as iron, folate, vitamins B12 and A); hemoglobinopathies, and infectious diseases such as malaria, tuberculosis, HIV, and hookworm) are the most common causes of anemia in children [5–7].

Anemia is a significant global public health concern that affects young children and pregnant women in particular [8, 9]. Globally, anemia affects 600 million children [10], with 43% of children under-five years of age estimated to be anemic [9, 11]. Based on the 2016 World Bank report, the prevalence in East African countries was ranged from 36% in Rwanda to 60% in Mozambique whereas in West African countries the prevalence was from 62% in Beni to 86% in Burkina Faso. Whereas, in South African countries, the prevalence of under-five children ranged from 37% in South Africa to 51% in Angola, and in Central Africa varied from 51% in Angola to 73% in Chad [12]. The prevalence varies across countries, with the highest-burden in sub-Saharan Africa (SSA) and South Asia [13]. It has significant short-and long-term consequences to the health of the children. Anemia has a detrimental impact on children's health, including developmental delay, reduced cognitive development (impaired learning and decreased school performance), low immunity, fatigue, difficulty of focusing, lethargy, increased mortality, and vulnerability to infection [14]. Moreover, childhood anemia is associated with decreasing the ability to fight infections and that causes significant morbidity and mortality in children [15].

The causes of anemia are multifactorial, and though infectious diseases and nutritional deficiencies are the leading causes of anemia in SSA [16, 17]. The finding of previous literature revealed that different factors are associated with anemia among children. These include: maternal age [18], twin births [19], birth order [20], residence [21], child age [22, 23], place of delivery [24, 25], deworming [26], childhood nutritional status [27], household wealth status

[18], maternal education [28], infectious diseases (malaria, hookworm) [29], and maternal anemia [30, 31]. Besides, in developing countries, anemia varies by socioeconomic factors [17, 32].

Despite the appreciable global progress in the socio-economic and health status of the community, sub-Saharan African countries are still faced with a huge number of under-five mortality [33]. About 67.6% of under-five children in Africa are suffering from anemia and responsible for 5–18% of under-five mortality. To achieve the targets of reducing child mortality of the Sustainable Development Goals (SDGs)-2030, it is necessary to generate adequate evidence on individual and community-level factors of anemia, which is highly crucial for the development of timely interventions in anemia prevention and treatment. There are studies conducted on the prevalence of anemia and its associated factors among children aged 6–59 months in sub-Saharan Africa. But these studies are unable to capture the ordinal nature of anemia status since the effects of anemia differ depending on the severity level of anemia (non-anemic, mild, moderate, and severe anemia). Therefore, we applied the multilevel ordinal logistic regression model to get a reliable estimate and avoid loss of information. This study has both public health and methodological significance. Regarding the public health perspective, this study was based on the pooled DHS data of 32 sub-Saharan African countries with very large sample size and this could increase the power of the study and the estimate can be generalized. Besides, the use of a multilevel approach can take into account the neighborhood effect, and the result can give the overall picture of SSA. Regarding the methodological perspective, as you can see previously published literature treat anemia as a binary outcome by categorizing no/yes but as you can understand treating mild, moderate, and severe anemia as yes is not statistically appropriate since there is a loss of information because the factor responsible for mild anemia may not be similar with the factor that can cause severe anemia.

## Methods

### Data source and sampling procedure

A secondary data analysis was done based on the most recent Demographic Health Survey (DHS) datasets conducted in 32 sub-Saharan African countries from 2005 to 2018. These datasets were appended together to investigate the prevalence and determinants of the severity of anemia among children aged 6–59 months in sub-Saharan Africa. The DHSs were a nationally representative survey that collects data on basic health indicators like mortality, morbidity, family planning service utilization, fertility, maternal and child health-related indicators. The data were derived from the measure DHS program and the detailed information about the surveys can be found in each countries DHS report. In the beginning, the country was stratified and selected in two stages. Each region/county of the country was stratified into urban and rural areas. Then, samples of Enumeration Areas (EAs) were selected independently in each stratum in two stages. Stratification and proportional allocation were achieved at each of the lower administrative levels by sorting the sampling frame within each sampling stratum before sample selection, according to administrative units in different levels, and by using a probability proportional to size selection at the first stage of sampling. In the first stage, Enumeration Areas (EAs) were randomly selected. In the second stage of selection, a fixed number of 20 to 28 households per cluster were selected with an equal probability systematic selection. Then, hemoglobin testing was carried out among children aged 6–59 months in the selected households using HemoCue rapid testing methodology. For the test, a drop of capillary blood was taken from a child's fingertip or heel and was drawn into the micro cuvette which was then analyzed using the photometer that displays the hemoglobin concentration. Then, anemic status was determined based on the hemoglobin level. Each country's survey consists of different

datasets including men, women, children, birth, and household datasets. For this study, we used the Kids Record dataset (KR file). Using the KR file we extract the dependent and independent variables for each country and then we append the data using the STATA command "append using". We pooled the DHS survey data of the 32 sub-Saharan Africa countries. For this study, a total weighted sample of 135,619 children aged 6–59 months was included.

## Study variables and measurements

**Dependent variable.**   The response variable of this study was the anemic status of children, which is an ordered categorical variable categorized into four ordinal categories; mildly anemic (hemoglobin level 10.0–10.9g/dl, moderately anemic (hemoglobin level 7.0–9.9g/dl), severely anemic (hemoglobin level <7.0g/dl), and not anemic (hemoglobin ≥11.0 g/dl). It was assessed based on the hemoglobin concentration in blood adjusted to the altitude.

In DHS, before determining a child is anemic or not, they take into account altitude. Then, they have adjusted, the hemoglobin adjustment was done by subtracting or adding the adjusted Hgb value to each individual observed Hgb value.

The Hgb adjustment was made using the formula;

$$= -0.0322 \, (\text{altitude} * 0.0032808) \text{ or}$$

$$+0.022 \, (\text{altitude} * 00032808)^2$$

The adjustment for altitude was done to take into account the reduction in oxygen saturation of the blood.

**Independent variables.**   Consistent with the study's objectives and given the hierarchical nature of DHS data where children and mothers were nested within the cluster, two levels of independent variables were considered. Individual-level factors considered were categorized as household-related characteristics, maternal-related and child-related characteristics. Household-related factors were household wealth status, the number of a household member, source of drinking water, sex of household head, and media exposure. Maternal related factors were maternal age, maternal education, marital status, maternal anemia, mothers Body Mass Index (BMI), the number of Antenatal Care (ANC) visits during pregnancy, place of delivery, Postnatal Care visit, taking an iron supplement during pregnancy, wanted birth, mothers current employment status, and maternal smoking status. Among child-related factors; the age of a child, size of child at birth, sex of a child, type of birth, birth order, diarrhea in the last two weeks, fever in the last two weeks, cough in the last two weeks, taking the drug for the intestinal parasite in the last six months, vitamin A supplementation in the last 6 months wasting status (Z-scores for Weight-for-Height (WHZ)), underweight status (Z-scores for Weight-for-Age (WAZ)) and stunting status (Z-scores for Height-for-Age (HAZ)). Level 2 (contained community-level variables) was the sub-Saharan African region and residence.

Stunting is defined as the children with height-for-age Z-score (HAZ) <−2SD, wasting is defined as the children with weight-for-height Z-score (WHZ) <−2SD and underweight is defined as the children with weight-for-age Z-score (WAZ) <−2SD. Maternal anemia was categorized into mild, moderate, and severe anemia for non-pregnant women was 10–11.9 g/dl, 7–9.9 g/dl, and <7 g/dl, respectively and for pregnant women, 10–10.9 g/dl, 7–9.9 g/dl, and <7 g/dl, respectively.

## Data management and analysis

The data were weighted using sampling weight, primary sampling unit, and strata before any statistical analysis to restore the representativeness of the survey and take into account the

sampling design when calculating standard errors, to get reliable statistical estimates. STATA version 14 statistical software was used for the data management and analysis. Owing to the ordinal nature of the outcome variable (non-anemic, mild, moderate, and severe anemic), a typical approach was the ordinal logistic regression model. To choose the appropriate ordinal model for the data, we have checked the Proportional Odds (PO) assumptions, which states that the effects of all independent variables are constant across categories of the outcome variable. After fitting the proportional odds model, the proportional odds assumption was tested using the Brant test. It tests the null hypothesis that there is no difference in the effects of independent variables across the levels of anemia. The Brant test revealed that the proportional odds assumption was fulfilled (p = 0.091). We, therefore, used the proportional odds model for assessing the association between anemia and independent variables.

Besides, the DHS data has hierarchical nature. Therefore, children and mothers nested within a cluster, and we assume that study subjects in the same cluster may share similar characteristics to participants in another cluster. This violates the independence observations and equal variance assumptions between clusters of the ordinal logistic regression model. This implies the need to take into account the heterogeneity between clusters by using an advanced model. Therefore, a multilevel proportional odds model was performed to get a reliable estimate and standard error.

Hence, since the Brant test was met, the multilevel proportional odds model gave a single Odds Ratio (OR) for an explanatory variable (severe vs moderate/mild/non-anemia, severe/moderate vs mild/non-anemia, and severe/moderate/mild vs non-anemic. Likelihood Ratio (LR) test, Intra-class Correlation Coefficient (ICC), and Median Odds Ratio (MOR) were computed to measure the variation of anemia across clusters. The ICC quantifies the degree of heterogeneity of anemia between clusters (the proportion of the total observed variation in anemia that is attributable to between cluster variations) [34].

$$ICC = \sigma^2/(\sigma^2 + \pi^2/3).$$

Where: the standard logit distribution has a variance of $\pi^2/_3$, $\sigma_\mu^2$ indicates the cluster variance.

The MOR quantifies the variation or heterogeneity in anemia between clusters in terms of odds ratio scale and is defined as the median value of the odds ratio between the cluster at high likelihood of anemia and cluster at lower risk when randomly picking out individuals from two clusters (EAs) [35].

$$MOR = exp^{\sqrt{(2*\partial 2*0.6745)}}, \ MOR = exp^{0.95*\partial}.$$

$\partial^2$ indicates that cluster-level variance

Four models were constructed for the multilevel logistic regression analysis. The first model was a null model without explanatory variables to determine the extent of cluster variation in childhood anemia. The second model was adjusted with individual-level variables; the third model was adjusted for community-level variables while the fourth was fitted with both individual and community level variables simultaneously. Model comparison was made based on deviance (-2Log-Likelihood Ratio (LLR)) since the models were nested models, and a model with the lowest deviance was the best-fitted model for the data.

Variables with a p-value ≤ 0.2 in the bi-variable multilevel proportional odds model were considered for the multivariable multilevel proportional odds model. In the multivariable multilevel proportional odds model, the Adjusted Odds Ratio (AOR) with 95% Confidence Interval (CI) were reported to declare the strength of association, and the statistical significance for the final model was set at p<0.05.

## Ethical consideration

As the study was a secondary data analysis of publicly accessible survey data from the MEA-SURE DHS program, this study did not require ethical approval and participant consent. We have granted permission from http:/www.dhsprogram.com to download and use the data for this study. In the datasets, there are no names of persons or household addresses recorded.

## Results

### Descriptive characteristics of the study participants

A total of 135,619 children aged 6–59 months were included (Table 1). Of these, 94,592 (59.3%) were from rural area, and 55,216 (40.6%) from East Africa, 51,332 (37.9%) from West Africa, 5,178 (3.8%) in Southern Africa and 23,892 (17.6%) in Central Africa. About 86,940 (64.1%) aged 24–59 months and 68,426 (50.5%) were males. The majority (67.4%) of the children got birth at a health facility, and 22,409 (16.5%) were small size at birth. Nearly half

**Table 1. Number of study participants for this study, and survey years.**

| Sub-Saharan region | Country | Sample size (n = 135,619) | Prevalence of anemia (%) | Study year |
|---|---|---|---|---|
| **East Africa** | Burundi | 5588 | 60.9 | 2016/17 |
| | Ethiopia | 8482 | 57.6 | 2016 |
| | Madagascar | 4845 | 51.2 | 2008/09 |
| | Malawi | 4677 | 63 | 2015/16 |
| | Mozambique | 4640 | 69.2 | 2011 |
| | Rwanda | 3283 | 36.6 | 2014/15 |
| | Tanzania | 7828 | 58.7 | 2015/16 |
| | Uganda | 3893 | 53.8 | 2016 |
| | Zambia | 7626 | 59 | 2018 |
| | Zimbabwe | 4354 | 38 | 2015 |
| **Southern Africa** | Lesotho | 1138 | 54.1 | 2014 |
| | Namibia | 1412 | 49.6 | 2013 |
| | Swaziland | 1813 | 44.4 | 2005 |
| | South Africa | 816 | 62 | 2016 |
| **West Africa** | Burkina Faso | 6043 | 88 | 2010 |
| | Benin | 5586 | 72 | 2018 |
| | Côte d'Ivoire | 2693 | 75.7 | 2011/2012 |
| | Ghana | 2312 | 66.8 | 2014 |
| | Gambia | 2829 | 71.5 | 2013 |
| | Guinea | 2973 | 75.3 | 2018 |
| | Mali | 3979 | 82.3 | 2018 |
| | Nigeria | 10222 | 68.1 | 2018 |
| | Niger | 4549 | 73.7 | 2012 |
| | Sierra leone | 4168 | 79.9 | 2013 |
| | Senegal | 3206 | 76.8 | 2010/2011 |
| | Togo | 2771 | 70.9 | 2013/2014 |
| **Central Africa** | Angola | 5221 | 65.3 | 2015 |
| | DR. Congo | 7164 | 60.1 | 2013/2014 |
| | Congo | 3395 | 67.1 | 2011/2012 |
| | Cameroon | 4190 | 58.2 | 2011 |
| | Gabon | 2513 | 61.3 | 2012 |
| | São Tomé e Príncipe | 1409 | 63.6 | 2008/09 |

(48.5%) of the children's mothers were aged 20–29 years, and 52,702 (38.9%) were born to mothers who did not attain formal education. About 21,666 (16.0%), and 30,264 (22.3%) had diarrhea and fever in the last two weeks, respectively. Regarding nutritional status, 35.8%, 16.9%, and 6.2% of the children were stunted, underweight and wasted, respectively (Table 2).

## Prevalence and severity of anemia in sub-Saharan Africa

The overall prevalence of anemia among children aged 6–59 months was 64.1% [95% CI: 63.8%, 64.4%]. This study showed that 26.2% [95% CI: 25.9%, 26.4%] of children aged 6–59 months had mild anemia, 34.9% [95% CI: 34.7%, 35.2%] moderate anemia and 3% [95% CI: 2.9%, 3.1%] severe anemia. The highest prevalence of anemia was found in children whose mothers were moderately, and severely anemic which was 76.8% and 76.7%, respectively. Regarding the severity of anemia, the highest prevalence of severe anemia was found in children whose mother was severely anemic (10.7%). Of the children born to mothers aged less than 20 years, 4.1%, 43.4%, and 26.4% of them were severely, moderately and mildly anemic, respectively (Table 3).

## Model fit statistics

The Brant test of parallel odds assumption showed that odds ratios appeared to have held constant across all cut-off points of childhood anemia status for the final model at a 5% significance level (p-value = 0.091). Therefore, the interpretations of odds ratio results obtained by modeling severely anemic vs moderately/mild/non-anemic; and anemic vs non-anemic were the same. In the null model, the ICC value was 12.73% [95% CI: 11.56%, 14.10], indicated that 12.73% of the total variability of level of anemia was due to differences between clusters while the remaining unexplained 87.27% of the total variability of level of anemia was attributable to the individual differences. Moreover, the MOR was 1.93 [95% CI: 1.86, 2.01] in the null model. The final model was the best-fitted model for the data since it has the lowest deviance value (Table 4).

## Individual and community-level determinants of anemia

To identify the determinants of anemia, the bivariable analysis was performed. Accordingly, maternal education, maternal age, household wealth status, family size, distance to the health facility, maternal anemia, media exposure, place of delivery, sex of the child, age of the child, size of child at birth, type of birth, taking drugs for an intestinal parasite, diarrhea, fever, birth order, wasting, stunting, underweight, residence and region were considered for the multivariable analysis(p<0.2). In the multivariable multilevel proportional odds model; maternal age, taking drugs for the intestinal parasite in the last six months and sub-Saharan African region were significantly associated with the lower odds of severity levels of anemia whereas maternal education, household wealth status, number of household members, maternal anemia, sex of the child, type of birth, fever in the last two weeks, age of the child, in the last two weeks, birth order, stunting, wasting, and underweight were significantly associated with higher odds of severity levels of anemia. The odds of having higher level of anemia among children whose mother aged 20–29 years, 30–39 years and 40–49 years were decreased by 18% [AOR = 0.82, 95% CI: 0.78, 0.86], 32% [AOR = 0.68, 95% CI: 0.65, 0.73] and 40% [AOR = 0.60, 95% CI: 0.56, 0.64] compared to children whose mother aged less than 20 years, respectively. Children whose mother education level at no formal education, primary education, and secondary education level had 1.73 times [AOR = 1.73, 95% CI: 1.60, 1.86], 1.39 times [AOR = 1.39, 95% CI: 1.29, 1.50], 1.27 times [AOR = 1.27, 95% CI: 1.18, 1.36] higher odds of a higher level of anemia than children whose mother had a higher level of education, respectively. Children from

**Table 2. Descriptive characteristics of children aged 6–59 months in sub-Saharan Africa.**

| Variable | Frequency (N = 135,619) | Percentage (%) |
|---|---|---|
| **Household characteristics** | | |
| **Household wealth status** | | |
| Poorest | 30,681 | 22.6 |
| Poorer | 29,744 | 21.9 |
| Middle | 27,421 | 20.2 |
| Richer | 26,224 | 19.3 |
| Richest | 21,548 | 15.9 |
| **Number of household members** | | |
| 1–4 | 33,048 | 24.4 |
| 5–8 | 70,468 | 52.0 |
| > 8 | 32,103 | 23.6 |
| **Source of drinking water supply** | | |
| Not improved | 38,218 | 28.2 |
| Improved | 97,400 | 71.8 |
| **Sex of household head** | | |
| Male | 108,320 | 79.9 |
| Female | 27,299 | 20.1 |
| **Media exposure** | | |
| No | 46,737 | 34.5 |
| Yes | 88,882 | 65.5 |
| **Maternal related characteristics** | | |
| **Maternal age (in year)** | | |
| <20 | 6,857 | 5.1 |
| 20–29 | 65,831 | 48.5 |
| 30–39 | 50,245 | 37.1 |
| ≥40 | 12,685 | 9.4 |
| **Maternal education** | | |
| No formal education | 52,702 | 38.9 |
| Primary | 47,371 | 34.9 |
| Secondary | 31,845 | 23.5 |
| Higher | 3,701 | 2.7 |
| **Marital status** | | |
| Never married | 7,500 | 5.5 |
| Married | 118,980 | 87.7 |
| Divorced/widowed | 9,139 | 6.7 |
| **Maternal anemia level** | | |
| Severe | 1,106 | 0.9 |
| Moderate | 17,369 | 13.4 |
| Mild | 36,617 | 28.3 |
| No anemia | 74,473 | 57.5 |
| **Maternal BMI** | | |
| Normal | 81,884 | 60.4 |
| Underweight | 12,503 | 9.2 |
| Overweight | 41,232 | 30.4 |
| **Number of ANC visit during pregnancy** | | |
| No | 52,101 | 38.4 |
| 1–3 | 30,651 | 22.6 |

(*Continued*)

**Table 2.** (Continued)

| Variable | Frequency (N = 135,619) | Percentage (%) |
| --- | --- | --- |
| ≥4 | 52,867 | 39.0 |
| **Place of delivery** | | |
| Home | 44,186 | 32.6 |
| Health facility | 91,433 | 67.4 |
| **PNC visit** | | |
| No | 98,742 | 72.8 |
| Yes | 36,877 | 27.2 |
| **Taking iron supplements during pregnancy** | | |
| No | 63,367 | 46.7 |
| Yes | 72,252 | 53.3 |
| **Wanted birth** | | |
| No | 9,659 | 7.1 |
| Yes | 125,960 | 92.9 |
| **Mothers current employment status** | | |
| Not working | 48,892 | 36.0 |
| Working | 86,727 | 64.0 |
| **Mother's smoking cigarette** | | |
| No | 134,566 | 99.2 |
| Yes | 1,053 | 0.8 |
| **Child's characteristics** | | |
| **Sex of child** | | |
| Male | 68,426 | 50.5 |
| Female | 67,193 | 49.5 |
| **Age of child (in months)** | | |
| 6–23 | 48,679 | 35.9 |
| 24–59 | 86,940 | 64.1 |
| **Size of child at birth** | | |
| Large | 46,547 | 34.3 |
| Average | 66,663 | 49.2 |
| Small | 22,409 | 16.5 |
| **Type of birth** | | |
| Single | 131,397 | 96.9 |
| Multiple | 4,222 | 3.1 |
| **Birth order** | | |
| 1 | 28,600 | 21.1 |
| 2–3 | 47,947 | 35.4 |
| 4–5 | 31,469 | 23.2 |
| ≥6 | 27,603 | 20.4 |
| **Diarrhea in the last two weeks** | | |
| No | 113,953 | 84.0 |
| Yes | 21,666 | 16.0 |
| **Cough in the last two weeks** | | |
| No | 105,653 | 77.9 |
| Yes | 29,966 | 22.1 |
| **Fever in the last two weeks** | | |
| No | 105,355 | 77.7 |
| Yes | 30,264 | 22.3 |

(*Continued*)

**Table 2.** (Continued)

| Variable | Frequency (N = 135,619) | Percentage (%) |
|---|---|---|
| **Taking drug for intestinal parasite in the last 6 months** | | |
| No | 77,194 | 56.9 |
| Yes | 58,425 | 43.1 |
| **Vitamin A supplementation in the last 6 months** | | |
| No | 54,981 | 40.5 |
| Yes | 80,637 | 59.5 |
| **Stunting status** | | |
| Normal | 87,037 | 64.2 |
| Stunted | 48,582 | 35.8 |
| **Underweight status** | | |
| Normal | 112,720 | 83.1 |
| Underweight | 22,899 | 16.9 |
| **Wasting status** | | |
| Normal | 127,183 | 93.8 |
| Wasted | 8,436 | 6.2 |
| **Community-level characteristics** | | |
| **Residence** | | |
| Rural | 94,592 | 69.3 |
| Urban | 41,027 | 30.3 |
| **Region** | | |
| East Africa | 55,216 | 40.7 |
| West Africa | 51,332 | 37.9 |
| Southern Africa | 5,178 | 3.8 |
| Central Africa | 23,892 | 17.6 |

BMI: Body Mass Index, ANC: Antenatal Care.

poorest, poorer, middle and richer household wealth were 1.39 times [AOR = 1.39, 95% CI: 1.33, 1.45], 1.32 times [AOR = 1.32, 95% CI: 1.26, 1.37], 1.20 times [AOR = 1.20, 95% CI: 1.15, 1.25], and 1.15 times [AOR = 1.15, 95% CI: 1.11, 1.20] higher odds of higher level of anemia compared to children from the richest household wealth, respectively. The odds of a higher level of anemia among children from the family size of 5–8 and >8 were 1.04 times [AOR = 1.04, 95% CI: 1.01, 1.06], 1.13 times [AOR = 1.13, 95% CI: 1.09, 1.16] higher than children from the family size of less than 5, respectively.

Children whose mother were mildly anemic, moderately anemic, and severely anemic were 1.54 times [AOR = 1.54, 95% CI: 1.51, 1.58], 1.93 times [AOR = 1.93, 95% CI: 1.87, 2.00] and 2.81 times [AOR = 2.81, 95% CI: 2.50, 3.16] higher odds of having a higher level of anemia than children whose mother were not anemic, respectively. Male children and multiple births had 1.13 times [AOR = 1.13, 95% CI: 1.11, 1.16] and 1.18 times [AOR = 1.18, 95% CI: 1.11, 1.25] higher odds of being at higher level anemia status compared to female children, and singletons, respectively. The odds of having a higher level of anemia among children aged 24–59 months were decreased by 54% [AOR = 0.46, 95% CI: 0.45, 0.47] compared to children aged 6–23 months. Children who were the 2nd-3rd, 4th-5th and 6th and above birth order were 1.07 times [AOR = 1.07, 95% CI: 1.04, 1.10], 1.14 times [AOR = 1.14, 95% CI: 1.10, 1.19] and 1.23 times [AOR = 1.23, 95% CI:1.17, 1.29] higher odds of having a higher level of anemia compared to first births, respectively.

Table 3.  The prevalence and severity of anemia based on the household related, community, child and maternal characteristics in sub-Saharan Africa.

| Variable | Categories | Anemia status and severity level (%) | | | | Overall anemia prevalence (%) |
|---|---|---|---|---|---|---|
| | | Severely anemic | Moderately anemic | Mildly anemic | Non-anemic | |
| Maternal age | <20 | 4.1 | 43.4 | 26.4 | 26.2 | 73.8 |
| | 20–29 | 3.2 | 35.9 | 26.2 | 34.7 | 65.3 |
| | 30–39 | 2.9 | 34.1 | 25.8 | 37.3 | 62.7 |
| | 40–49 | 2.8 | 32.2 | 26.2 | 38.8 | 61.2 |
| Maternal education | No formal education | 4.8 | 42.8 | 25.2 | 27.2 | 72.8 |
| | Primary | 2.3 | 32.0 | 26.3 | 39.5 | 60.5 |
| | Secondary | 1.7 | 29.3 | 27.2 | 41.8 | 58.2 |
| | Higher | 1.0 | 20.1 | 26.4 | 52.6 | 47.5 |
| Residence | Urban | 1.8 | 31.6 | 27.6 | 39.0 | 60.8 |
| | Rural | 3.5 | 36.4 | 25.6 | 34.5 | 65.5 |
| Household wealth status | Poorest | 4.4 | 39.2 | 25.7 | 30.7 | 69.3 |
| | Poorer | 3.6 | 37.5 | 25.7 | 33.3 | 66.8 |
| | Middle | 2.9 | 35.7 | 26.8 | 35.3 | 64.7 |
| | Richer | 2.3 | 32.8 | 26.7 | 38.2 | 61.8 |
| | Richest | 1.3 | 26.9 | 26.9 | 44.8 | 65.2 |
| sub-Saharan Africa region | East Africa | 2.1 | 28.1 | 25.8 | 44.0 | 56.0 |
| | South Africa | 1.4 | 25.4 | 23.9 | 49.3 | 50.7 |
| | West Africa | 4.6 | 45.0 | 25.6 | 24.9 | 75.1 |
| | Central Africa | 2.1 | 31.4 | 28.7 | 37.8 | 62.2 |
| Maternal anemia | Non-anaemic | 2.1 | 29.8 | 25.6 | 42.5 | 57.5 |
| | Mild | 3.7 | 40.2 | 27.1 | 29.0 | 71.0 |
| | Moderate | 5.2 | 46.0 | 25.6 | 23.2 | 76.8 |
| | Severe | 10.7 | 45.0 | 21.0 | 23.3 | 76.7 |
| Sex of child | Male | 3.3 | 36.1 | 25.9 | 34.8 | 65.3 |
| | Female | 2.8 | 33.8 | 26.4 | 37.1 | 63.0 |
| Type of birth | Single | 3.0 | 34.8 | 26.2 | 36.0 | 64.0 |
| | Multiple | 4.6 | 38.5 | 24.9 | 32.0 | 68.0 |
| Birth order | 1 | 2.9 | 33.3 | 26.1 | 37.7 | 62.3 |
| | 2–3 | 2.8 | 33.9 | 26.1 | 37.2 | 62.8 |
| | 4–5 | 3.0 | 35.8 | 26.9 | 34.4 | 65.6 |
| | ≥6 | 3.5 | 37.4 | 25.6 | 33.5 | 66.5 |
| Age of child (in months) | 6–23 | 4.2 | 45.2 | 26.7 | 23.9 | 76.1 |
| | 24–59 | 2.5 | 29.7 | 25.7 | 42.1 | 57.9 |
| Stunting status | Normal | 2.5 | 33.2 | 26.4 | 38.0 | 62.0 |
| | Stunted | 4.2 | 39.1 | 25.4 | 31.3 | 68.7 |
| Wasting status | Normal | 2.8 | 34.7 | 26.2 | 36.3 | 63.8 |
| | Wasted | 6.8 | 43.4 | 24.1 | 25.7 | 74.3 |
| Underweight status | Normal | 2.5 | 33.7 | 26.4 | 37.4 | 62.6 |
| | Underweight | 6.2 | 43.1 | 24.2 | 26.5 | 73.5 |
| Size of child at birth | Average | 3.0 | 34.9 | 25.9 | 36.2 | 63.8 |
| | Small | 3.9 | 37.2 | 25.3 | 33.6 | 66.5 |
| | Large | 2.9 | 34.8 | 26.6 | 35.8 | 64.3 |
| Diarrhea in the last two weeks | No | 2.8 | 34.0 | 26.2 | 36.9 | 63.1 |
| | Yes | 4.4 | 41.9 | 25.2 | 28.5 | 71.5 |
| Fever in the last two weeks | No | 2.5 | 33.3 | 26.5 | 37.7 | 62.4 |
| | Yes | 5.2 | 42.0 | 24.4 | 28.4 | 71.6 |
| Overall prevalence (95% CI) | | 3 [2.9, 3.1] | 34.9[34.7, 35.2] | 26.2 [25.9, 26.4] | 35.9 [35.6, 36.1] | 64.1 [63.8, 64.4] |

**Table 4. Individual and community-level factors associated with anemia among children aged 6–59 months in sub-Saharan Africa.**

| Variables | Null model | Model 1 | Model 2 | Model 3 |
|---|---|---|---|---|
| | | AOR with 95% CI | AOR with 95% CI | AOR with 95% CI |
| **Individual-level variables** | | | | |
| **Maternal age (in years)** | | | | |
| <20 | | 1 | | 1 |
| 20–29 | | 0.82 [0.78, 0.87] | | 0.82 [0.78, 0.86]** |
| 30–39 | | 0.69 [0.65, 0.73] | | 0.68 [0.65, 0.73]** |
| 40–49 | | 0.60 [0.56, 0.64] | | 0.60 [0.56, 0.64]** |
| **Maternal education** | | | | |
| No formal education | | 2.10 [1.95, 2.25] | | 1.73 [1.60, 1.86]* |
| Primary | | 1.27 [1.18, 1.37] | | 1.39 [1.29, 1.50]* |
| Secondary | | 1.21 [1.13, 1.30] | | 1.27 [1.18, 1.36]* |
| Higher | | 1 | | 1 |
| **Household wealth status** | | | | |
| Poorest | | 1.31 [1.26, 1.36] | | 1.39 [1.33, 1.45]** |
| Poorer | | 1.28 [1.23, 1.33] | | 1.32 [1.26, 1.37]* |
| Middle | | 1.19 [1.15, 1.24] | | 1.20 [1.15, 1.25]* |
| Richer | | 1.15 [1.11, 1.19] | | 1.15 [1.11, 1.20]* |
| Richest | | 1 | | 1 |
| **Family size** | | | | |
| ≤4 | | 1 | | 1 |
| 5–8 | | 1.05 [1.02, 1.08] | | 1.04 [1.01, 1.06]* |
| ≥9 | | 1.25 [1.21, 1.30] | | 1.13 [1.09, 1.16]* |
| **Maternal anemia** | | | | |
| No-anemic | | 1 | | 1 |
| Mild | | 1.62 [1.58, 1.66] | | 1.54 [1.51, 1.58]* |
| Moderate | | 2.20 [2.13, 2.27] | | 1.93 [1.87, 2.00]* |
| Severe | | 3.06 [2.73, 3.44] | | 2.81 [2.50, 3.16]** |
| **Media exposure** | | | | |
| No | | 1 | | 1 |
| Yes | | 1.02 [0.99, 1.04] | | 0.94 [0.92, 1.01] |
| **Place of delivery** | | | | |
| Home | | 1 | | 1 |
| Health facility | | 0.99 [0.97, 1.02] | | 1.04 [0.98, 1.07] |
| **Sex of child** | | | | |
| Male | | 1.13 [1.11, 1.16] | | 1.13 [1.11, 1.16]** |
| Female | | 1 | | 1 |
| **Age of child (in months)** | | | | |
| 6–23 | | 1 | | 1 |
| 24–59 | | 0.47 [0.46, 0.48] | | 0.46 [0.45, 0.47]* |
| **Size of child at birth** | | | | |
| Average | | 1 | | 1 |
| Small | | 0.99 [0.97, 1.02] | | 0.99 [0.97, 1.03] |
| Large | | 1.01 [0.99, 1.04] | | 0.98 [0.96, 1.01] |
| **Type of birth** | | | | |
| Single | | 1 | | 1 |
| Multiple | | 1.20 [1.13, 1.28] | | 1.18 [1.11, 1.25]** |
| **Taking drug for intestinal parasite in the last 6 months** | | | | |

*(Continued)*

**Table 4.** (Continued)

| Variables | Null model | Model 1 | Model 2 | Model 3 |
|---|---|---|---|---|
| | | AOR with 95% CI | AOR with 95% CI | AOR with 95% CI |
| No | | 1 | | 1 |
| Yes | | 0.87 [0.85, 0.89] | | 0.91 [0.89, 0.93]** |
| **Diarrhea in the last two weeks** | | | | |
| No | | 1 | | 1 |
| Yes | | 1.11 [1.08, 1.14] | | 1.12 [1.09, 1.16]* |
| **Fever in the last two weeks** | | | | |
| No | | 1 | | 1 |
| Yes | | 1.45 [1.40, 1.46] | | 1.46 [1.42, 1.49]* |
| **Birth order** | | | | |
| 1 | | 1 | | 1 |
| 2–3 | | 1.06 [1.03, 1.10] | | 1.07 [1.04, 1.10]* |
| 4–5 | | 1.13 [1.08, 1.17] | | 1.14 [1.10, 1.19]** |
| ≥6 | | 1.18 [1.12, 1.23] | | 1.23 [1.17, 1.29]** |
| **Wasting status** | | | | |
| Normal | | 1 | | 1 |
| Wasted | | 1.14 [1.08, 1.19] | | 1.09 [1.04, 1.15]* |
| **Underweight status** | | | | |
| Normal | | 1 | | 1 |
| Underweight | | 1.30 [1.25, 1.34] | | 1.24 [1.20, 1.28]** |
| **Stunting** | | | | |
| Normal | | 1 | | 1 |
| Stunted | | 1.23 [1.20, 1.26] | | 1.29 [1.26, 1.32]** |
| **Community level variable** | | | | |
| **Residence** | | | | |
| Rural | | | 1.41 [1.38, 1.44] | 1.02 [0.99, 1.05] |
| Urban | | | 1 | 1 |
| **sub-Saharan African region** | | | | |
| West Africa | | | 1 | 1 |
| East Africa | | | 0.40 [0.39, 0.41] | 0.48 [0.47, 0.49]* |
| Central Africa | | | 0.34 [0.32, 0.36] | 0.59 [0.57, 0.61]* |
| South Africa | | | 0.56 [0.55, 0.58] | 0.45 [0.43, 0.48]* |
| **/cut1** | -0.60 [-0.61, -0.58] | -0.17 [-0.26,-0.08] | -0.89[-0.91,-0.86] | -0.64 [-0.74, -0.55] |
| **/cut2** | 0.48 [0.46, 0.50] | 1.01 [0.92, 1.11] | 0.23 [0.21, 0.26] | 0.57 [0.47, 0.66] |
| **/cut3** | 3.46 [3.43, 3.49] | 4.19 [4.09, 4.28] | 3.28 [3.25, 3.32] | 3.77 [3.67, 3.87] |
| **Random effect analysis result** | | | | |
| Community level variance | 0.48 [0.43, 0.54] | 0.023 [0.019, 0.03] | 0.029 [0.024, 0.037] | 0.022 [0.017, 0.027] |
| LR-test | Prob > = chibar2 <0.001 | | | |
| ICC | 12.73% [11.56%, 14.10] | | | |
| MOR | 1.93 [1.86, 2.01] | | | |
| LLR | -162528.7 | -158892.5 | -146190.1 | -144602.4 |
| Deviance (-2LLR) | 325057.4 | 317785 | 292380.2 | 289204.8 |

*p-value<0.05

**p-value<0.01: AOR = Adjusted Odds Ratio: CI: Confidence Interval: ICC = Intra-class Correlation Coefficient: LR = Likelihood Ratio: LLR = Log-likelihood Ratio: MOR: Median Odds Ratio: WAZ = Z-scores for Weight-for-Age: WHZ = Weight-for-Height: HAZ: Height-for-Age.

The odds of being at higher anemia status among children who took drugs for the intestinal parasite in the last six months were decreased by 9% [AOR = 0.91, 95% CI: 0.89, 0.93] than those who did not take drugs. Children who had diarrhea and fever in the last two weeks had 1.12 times [AOR = 1.12, 95% CI: 1.09, 1.16], and 1.46 times [AOR = 1.46, 95% CI: 1.42, 1.49] higher odds of a higher level of anemia compared to children who did not have diarrhea and fever, respectively. Stunted, wasted and underweight children had 1.29 times [AOR = 1.29, 95% CI: 1.26, 1.32], 1.09 times [AOR = 1.09, 95% CI: 1.04, 1.15], and 1.24 times [AOR = 1.24, 95% CI: 1.20, 1.28] higher odds of higher level of anemia, respectively. The odds of being at higher level of anemia among children in East Africa, Central Africa and South Africa were decreased by 52% [AOR = 0.48, 95% CI: 0.47, 0.49], 41% [AOR = 0.59, 95% CI: 0.57, 0.61], and 55% [AOR = 0.45, 95% CI: 0.43, 0.48] compared to children in West Africa, respectively (Table 4).

## Discussion

The prevalence of anemia among children in sub-Saharan Africa was 64.11% [95% CI: 63.85%, 64.36%], which revealed anemia among children is still a major public health problem in sub-Saharan Africa. Even though the combined strategies particularly iron supplementation and infectious disease management (such as malaria and helminth infections) are being introduced by the WHO to combat anemia, anemia remains a serious health care problem in sub-Saharan Africa. It is higher than the prevalence reported in Brazil [36], Europe [37, 38], and Ecuador [39]. The potential reason may be due to the long-standing prevalence of severe malnutrition among under-five children, because of insufficient dietary intake of nutrients, in sub-Saharan Africa [40, 41]. Besides, sub-Saharan African children are highly affected by infectious diseases such as malaria, hookworms, Schistosoma, and visceral leishmaniasis, due to their frequent exposure to poor sanitation and environmental conditions that favor the transmission and spread of parasites [42–44].

Furthermore, in the final model, we found that maternal age, taking drugs for the intestinal parasite in the last six months, and sub-Saharan African region were significantly associated with the lower odds of severity levels of anemia. Whereas maternal education, household wealth status, number of household members, maternal anemia, sex of the child, type of birth, fever in the last two weeks, age of the child, diarrhea in the last two weeks, birth order, stunting, wasting, and underweight were significantly associated with higher odds of severity levels of anemia.

Poor maternal education was significantly associated with increased odds of childhood anemia. It is consistent with studies reported in Korea [28] and Mexico [45], it may be because maternal education is a good indicator for nutritional outcomes of children [46, 47]. Maternal education contributes to raising the maternal understanding about infant health and nutrition (such as exclusive breastfeeding and appropriate complementary feeding), which in turn contributes to enhancing the quality of diets consumed by children [48]. Besides, mothers' level of education can positively influence practices related to the health care and feeding practice of their children [49]. Children born to mothers aged less than 20 years had higher odds of a higher level of anemia compared to children born to mothers aged 20 years and above. This was consistent with studies reported in Tanzania [50] and low- and middle-income countries [51]. It could be due to babies born to younger mothers are more likely to be a preterm and low birth weight that results in the newborn prone to neonatal infections and malnutrition, these could increase their risk of anemia [52]. Furthermore, it may also predispose to deficiencies of hematopoiesis-related nutrients due to the inexperience of adolescent mothers with good infant feeding practices and prevention of infections and infestations [53].

In this study, children from families with low household wealth had increased odds of higher levels of anemia than children from rich households. It is consistent with previous studies reported in Bangladesh [54], India [30], and Nepal [55]. The possible explanation might be because poverty is strongly associated with food insecurity and hence children from households with low wealth status may not have an access to foods rich in iron, vitamin B12, and folic acid, which in turn increases their risk of developing anemia [56]. The other possible explanation is that families with low income are less likely to purchase nutrient-rich foods, secure food availability, and afford health services during illness for their children. Children from large family size had increased odds of a higher level of anemia than children from smaller family size. It is supported by previous studies in Egypt [57] and China [58]. It is because children from a large family size may not get adequate nutrition than children from small size families. So, inadequate intake of nutrients such as iron, folate, and vitamin B12 increases the risk of anemia among children [59].

Maternal anemia was significantly associated with higher odds of higher levels of anemia among children. It is in line with study findings in Southern Africa [60], Brazil [61], and Bangladesh [21]. This may be due to the mother is the primary source of food for children and the children share a similar diet, so their eating habits and quality of life could be identical [61, 62]. Also, through transplacental transmission and breastfeeding, infectious causes of anemia such as malaria and HIV/AIDS that can interfere with their development of red blood cells and iron stores may be transmitted to their infants [63–65]. In addition, anemic mothers may not have adequate iron, zinc, folate, and vitamin B12 in their breast milk, which could make the child vulnerable to anemia [66].

The study at hand also revealed that male children had higher odds of a higher level of anemia compared to female children. This is in line with studies in low-income countries [67] and Bangladesh [21]. It might be due to the rapid growth and development of male children in the first few years of life [68] that could increase their micronutrient demands, including vitamin A, folate, and iron to increase their risk of anemia compared to female children. Children aged 24–59 months had higher odds of a higher level of anemia compared to children aged 6–23 months. It was consistent with studies reported in India [18] and Bangladesh [21]. It could be since the age of 6–23 months is the critical period to initiate complementary feeding and for exposure to contaminated food and water, which might increase the incidence of intestinal infections such as typhoid, amoebiasis, giardiasis, ascariasis, and hookworm infections [69].

Children with a history of diarrhea and fever had higher odds of higher levels of anemia compared to children who did not have diarrhea and fever, respectively. It is consistent with studies reported in Southern Africa [60] and Indonesia [70]. This could be due to children with febrile and diarrheal illness might have a loss of appetite, and decreased absorption of necessary nutrients (iron, folate, and vitamin B12) that might increase the likelihood of anemia [71]. Besides, the presence of diarrhea and fever might indicate the presence of infectious diseases such as visceral leishmaniasis, malaria, hookworm, ascariasis, giardiasis, and amoebiasis, which are the leading causes of anemia in children [72, 73].

In this study, stunted, wasted, and underweight children had higher odds of a higher level of anemia compared to normal children. This was consistent with studies reported in India [74], China [75], and Brazil [76]. Aside from the deprivation of nutrients required for haematopoiesis, poor nutritional status is associated with poor immunity and, therefore, infections and infestations also have synergistic effects of micronutrient deficiencies for causing anemia [77]. Besides, undernourished children are prone to micronutrient deficiencies, such as iron, vitamin A, vitamin B12, and folic acid, which are helpful for haemoglobin and DNA synthesis during red blood cell production, and in turn, results in anemia [77].

Children who took drugs for intestinal parasites in the last 6 months had lower odds of a higher level of anemia compared to children who did not take drugs for intestinal parasites. It is consistent with sub-Saharan Africa [78] and Thailand [79]. This could be because anemia can result from intestinal parasites and hence taking drugs for intestinal parasites can decrease the risk of having anemia in children [80]. Multiple births are at higher risk of a higher level of anemia compared to singletons. The possible explanation is due to multiple births are more likely to be born prematurely, have low birth weight, and are at higher risk of malnutrition than single infants, which could increase the risk of experiencing anemia [81, 82]. Children with two or more older siblings in the household had higher odds of a higher level of anemia than first birth. It is consistent with studies reported in Central India [20] and Egypt [57]. The possible explanation could be due to the reason that increasing birth order might be related to the depletion of nutrients such as iron, folate, and vitamin B12 in the mother and this could result in anemia among children [83]. This could indicate poor access to maternal health care services such as ANC services and nutritional supplementations among multiparous mothers [84].

This study has strengths and limitations. This study was based on a pooled nationally representative DHS survey of the 32 sub-Saharan African countries. In addition, the data was weighted and a multilevel ordinal logistic regression analysis was done to get a reliable estimate and standard error. Besides, this study was based on a large sample size that had adequate power to detect the true effect of the independent variables. As a limitation, since the study used cross-sectional data, we cannot establish a causal relationship between anemia and the identified independent variables. In addition, since this study was based on secondary data, we were not able to investigate all factors that may be relevant to anemia in children, including eating habits, parasite infestations (malaria, Visceral Leishmaniasis, and hookworm), previous hospitalization, and use of nutritional supplements (such as vitamin B12 and folate). Moreover, variables such as the birth size of a child which is the subjective measurement of the birth size of a child were included in this study since the measured birth weight was not found for the majority of the children so this might overestimate or underestimate the effect size of birth size.

## Conclusions

In conclusion, anemia among children aged 6–59 months in sub-Saharan Africa was a major public health problem. Sex of child, maternal age, maternal education, type of birth, fever in the last two weeks, diarrhea in the last two weeks, taking drugs for an intestinal parasite, stunted, wasted, underweight, child age, birth order, household wealth status, family size, maternal anemia, and sub-Saharan African region was found significant determinants of the severity level of anemia. Improving access to education, providing drugs for an intestinal parasite, interventions to address maternal anemia, febrile illness, and diarrheal disease, and strengthening the economic status of the family are recommended to reduce childhood anemia. Furthermore, it is better to strengthen the strategies of early detection and management of stunted, wasted, and underweight children to decrease child anemia.

## Acknowledgments

We would like to thank the measure DHS program for providing the datasets.

## Author Contributions

**Conceptualization:** Getayeneh Antehunegn Tesema, Misganaw Gebrie Worku, Zemenu Tadesse Tessema, Achamyeleh Birhanu Teshale, Adugnaw Zeleke Alem, Yigizie Yeshaw, Tesfa Sewunet Alamneh, Alemneh Mekuriaw Liyew.

**Data curation:** Getayeneh Antehunegn Tesema, Misganaw Gebrie Worku, Zemenu Tadesse Tessema, Achamyeleh Birhanu Teshale, Adugnaw Zeleke Alem, Yigizie Yeshaw, Tesfa Sewunet Alamneh, Alemneh Mekuriaw Liyew.

**Formal analysis:** Getayeneh Antehunegn Tesema, Misganaw Gebrie Worku, Zemenu Tadesse Tessema, Achamyeleh Birhanu Teshale, Adugnaw Zeleke Alem, Yigizie Yeshaw, Tesfa Sewunet Alamneh, Alemneh Mekuriaw Liyew.

**Investigation:** Getayeneh Antehunegn Tesema, Misganaw Gebrie Worku, Zemenu Tadesse Tessema, Achamyeleh Birhanu Teshale, Adugnaw Zeleke Alem, Yigizie Yeshaw, Tesfa Sewunet Alamneh, Alemneh Mekuriaw Liyew.

**Methodology:** Getayeneh Antehunegn Tesema, Misganaw Gebrie Worku, Zemenu Tadesse Tessema, Achamyeleh Birhanu Teshale, Adugnaw Zeleke Alem, Yigizie Yeshaw, Tesfa Sewunet Alamneh, Alemneh Mekuriaw Liyew.

**Software:** Getayeneh Antehunegn Tesema, Misganaw Gebrie Worku, Zemenu Tadesse Tessema, Achamyeleh Birhanu Teshale, Adugnaw Zeleke Alem, Yigizie Yeshaw, Tesfa Sewunet Alamneh, Alemneh Mekuriaw Liyew.

**Validation:** Getayeneh Antehunegn Tesema, Misganaw Gebrie Worku, Zemenu Tadesse Tessema, Achamyeleh Birhanu Teshale, Adugnaw Zeleke Alem, Yigizie Yeshaw, Tesfa Sewunet Alamneh, Alemneh Mekuriaw Liyew.

**Visualization:** Getayeneh Antehunegn Tesema, Misganaw Gebrie Worku, Zemenu Tadesse Tessema, Achamyeleh Birhanu Teshale, Adugnaw Zeleke Alem, Yigizie Yeshaw, Tesfa Sewunet Alamneh, Alemneh Mekuriaw Liyew.

**Writing – original draft:** Getayeneh Antehunegn Tesema, Misganaw Gebrie Worku, Zemenu Tadesse Tessema, Achamyeleh Birhanu Teshale, Adugnaw Zeleke Alem, Yigizie Yeshaw, Tesfa Sewunet Alamneh, Alemneh Mekuriaw Liyew.

**Writing – review & editing:** Getayeneh Antehunegn Tesema, Misganaw Gebrie Worku, Zemenu Tadesse Tessema, Achamyeleh Birhanu Teshale, Adugnaw Zeleke Alem, Yigizie Yeshaw, Tesfa Sewunet Alamneh, Alemneh Mekuriaw Liyew.

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
