## [Decision Letter · Decision Letter 0]

31 Dec 2020

PONE-D-20-37075

Prevalence and determinants of severity levels of anaemia among children aged 6-59 months in sub-Saharan Africa: a multilevel ordinal logistic regression analysis

PLOS ONE

Dear Dr. Tesema,

Thank you for submitting your manuscript to PLOS ONE. After careful consideration, we feel that it has merit but does not fully meet PLOS ONE’s publication criteria as it currently stands. Therefore, we invite you to submit a revised version of the manuscript that addresses the points raised during the review process.

Four experts in the field handled your manuscript, and we are grateful for their time and contributions. Although interest was found in your study, several major concerns arose that require to attention. Please address ALL of the reviewers' comments in your revised manuscript.

We look forward to receiving your revised manuscript.

Kind regards,

Frank T. Spradley

Academic Editor

PLOS ONE

2.We noticed you have some minor occurrence of overlapping text with the following previous publication(s), which needs to be addressed:

https://www.hindawi.com/journals/bmri/2020/6907395/

In your revision ensure you cite all your sources (including your own works), and quote or rephrase any duplicated text outside the methods section. Further consideration is dependent on these concerns being addressed.

Reviewers' comments:

Reviewer's Responses to Questions

**Comments to the Author**

1. Is the manuscript technically sound, and do the data support the conclusions?

Reviewer #1: Yes

Reviewer #2: Yes

Reviewer #3: Partly

Reviewer #4: Yes

2. Has the statistical analysis been performed appropriately and rigorously? 

Reviewer #1: Yes

Reviewer #2: Yes

Reviewer #3: Yes

Reviewer #4: Yes

3. Have the authors made all data underlying the findings in their manuscript fully available?

Reviewer #1: Yes

Reviewer #2: Yes

Reviewer #3: Yes

Reviewer #4: Yes

4. Is the manuscript presented in an intelligible fashion and written in standard English?

Reviewer #1: No

Reviewer #2: Yes

Reviewer #3: No

Reviewer #4: No

5. Review Comments to the Author

Reviewer #1: PONE-D-20-37075

The manuscript requires intense editing for spellings and grammar.

INTRODUCTION

This is deficient in the previously reported prevalence rates of childhood anaemia in different African populations. This is needed to establish the burden of anaemia in African children. The burden of anaemia in childhood should be filtered from childhood mortality studies in Africa. All these are required to justify the present study.

Lines 62-63: This is not totally relevant to childhood.

Line 86: What proportion of childhood mortality is related to anaemia?

Line 90: What “problem” and what “region”?

Lines 95-96: What is the gap in knowledge the study proposed to fill?

METHOD

Secondary data analysis. Detailed methodology.

Line 109: What does (46) represent? If it is a reference, it is inappropriate to jump from reference #32 to reference #46. Even the item #46 on the list of references does not tally with the context within the text.

RESULTS

Line 198: “About” 21666? Figures should be exact.

Lines 132 and 200: What defined which nutritional status?

Lines: 204-206: Prevalence rates should be to a single decimal place.

Line 217: “cut of” or “cut-off”

DISCUSSION

Lines 295/298: Not specific about which factor increased or decreased with the possible risk of anaemia

Line 313: Inexperience about good infant feeding practices and prevention of infections and infestations may also predispose to deficiencies of haematopoiesis-related nutrients.

Line 339: Repetition of “vitamin A”

Lines 342-344: Diarrhea in 6-23 month age group will not explain the higher odd of higher level of anaemia in the 24-59 month age group.

Lines 345-347: The role of diarrhoea in childhood anaemia could only be justified if it is frequently recurrent or protracted to cause nutrient losses and nutrient store depletion.

Lines 349-351: Give examples of intestinal parasitosis known to cause fever, diarrhoea and anaemia in children.

Lines 352-354: Not clear why wasted children were separated from stunted and underweight children since they are all at risk of higher level of anaemia.

Lines 255-359: Aside deprivation of nutrients required for haematopoiesis, poor nutrition is also associated with poor immunity. Therefore infections and infestations also compound the effects of micronutrient deficiencies.

Reviewer #2: Anaemia has been a public health burden among under-5 children in sub-Saharan African countries. Therefore, in trying to understand the prevalence and determinants of severity levels of anaemia among children aged 6-59 months in sub-Saharan Africa and ways of reducing the burden, I found merit in the paper.

The paper is well written, the issues tackled are relevant, the methodology applied is appropriate, and the results are properly written. However, the authors were not consistent with using the words anaemia and anemia, sub-Sharan, and Sub-Saharan. Therefore, the entire manuscript requires thorough proofreading and editing.

In lines 51, 66, 67, 68, 69, 71, 75, 78, and 79 of pages 3-4, anemia should be changed to anaemia. Please be consistent and edit the entire paper.

In lines 70, 86, and 92 of page 3, sub-Saharan Africa should be written as sub-Saharan Africa. Please check the entire paper.

Methods

In line 103 of page 4, the authors should refer the readers to Table 1 that shows the list of countries involved in the study. In lines 104 and 106, change Sub to sub. There was no elaboration on the study population and sample size. The inclusion and exclusion criteria were also missing. Highlight information got from the DHS. The paper should also explain how the datasets from the surveys were merged.

The authors failed to list the percentage distribution of anaemia in each of the 32 countries.

In line 23 of page 2, the authors wrote anaemia and later changed it to anemia in line 26 of page 2. Hence, they continued interchanging the two words. Thorough editing is highly required.

The -2LLN acronym appeared first in line 179 of page 8; therefore, the authors should write the full meaning at the location in the article.

On page 7, the authors should clearly explain the analytical strategy and process, providing references.

Results and discussion

In line 194 of page 9, the authors mention the percentage distribution of anaemia among children in East Africa. It is also necessary that the percentage distribution of childhood anaemia in West Africa, Central Africa, and Southern Africa should be mentioned.

In line 202 of page 10, change Sub to sub.

In lines 217, 219, 221, 222, and 227 of page 10, change anemia to anaemia. Please be consistent and edit the whole manuscript.

In line 272 of page 12, South Africa should be changed to Southern Africa.

Correct the anemia in line 279 and Sub-Sharan Africa in line 280 of page 12.

A table of unadjusted ORs is missing.

Lines 287, 290, 291, 296, and 307 of page 13 should also be corrected.

Correct the anemia in lines 326 and 327 of page 14.

Lines 334 and 335 of page 15 should also be corrected.

Correct the anemia in lines 372 and 381 of page 16.

Reviewer #3: Manuscript Number: PONE-D-20-37075

Prevalence and determinants of severity levels of anaemia among children aged 6-59 months in sub-Saharan Africa: a multilevel ordinal logistic regression analysis

Feedback to authors

This article thought to identify a one figure for anaemia prevalence for under five years old children in Sub-Saharan Africa disaggregated by severity levels (mild, moderate, and severe) as well as to identify its determinants considering the severity levels factor. The author pooled data from previously conducted DHS in Africa and analyzed it using a multilevel ordinal logistic regression.

In general, the topic is good and the analysis design was nice to address the research question. However, there are some points to be taken into consideration to make the manuscript clearer and better designed.

Abstract:

Background:

In page 2, lines 26-27; the sentence “Though anemia is preventable, it remains ...” looks odd. It is distracting from the main scope of the study. Is better to be removed.

There is no valid justification stated for this research. Thus, the word “therefore” in the sentence “Therefore, this study ...” (in Page 2, line 27) seems strange. It is advised either to give reasonable justification or to remove it.

Methods:

In page 2, line 30; “This study was based on the most recent Demographic and Health Survey data ...” what is the limit/definition of recent. How far recent Demographic and Health Survey data are accepted to be included in this study. Please define.

In page 2, lines 32-33; “... and the ordinal nature of anaemia”. What is this ordinal nature? “... a multilevel ordinal logistic regression model was applied.” What is the study outcome?

In page 2, lines 33-34; “Proportional odds assumption was tested by Brant test ...” what is the purpose of using this test?

In page 2, lines 36; “... since the models were nested models.” This part of the sentence is better be deleted.

Independent variables were not described. You don’t need to list them, but only describe them in summary; therefore, the reader could have an idea about what you are looking at.

Results:

In page 2, lines 40-41; “... prevalence of anaemia among children aged 6-59 months in sub-Saharan Africa...” in the results, use your study population and do not apply your results to the general population. So, replace sub-Saharan Africa by another expression/word to reflect your own findings.

In page 2, lines 45; “... were significantly associated with an increased odd of higher...” correct the grammar of “an increased odd” to “increased odds”

While the authors used an ordinal logistic regression, nothing is mentioned about the order of severity in the results?!!!

Conclusion:

Nothing is mentioned about the level of severity in the conclusion.

The main article:

Background:

In page 3, lines 49-50; “... World Health 60 Organization (WHO) defines childhood anemia as a Hb concentration below 110 g/L.” The WHO has three cut-off levels for defining anaemia in childhood depending on age group. Please revise and correct your statement. Ref. [3] is not a WHO publication. If you are referring your statement to the WHO as a reference organization for setting standard definitions, then you should cite reference from that organization itself and not what others stated what WHO had said! Please revise.

In page 3, lines 62-66; the author stated many causes that he did not studied. Stating such issues in the background without being related to the study seems redundancy and could distract the reader from the target of the study. Consider deleting it.

In page 3, lines 64-65; the word “hemoglobinopathies” is duplicated

In page 4, lines 79-84; the author mentioned - from previous studies - many factors associated with anaemia; however, some of them (infectious diseases (malaria, hookworm)) were not included in this study! Explain why.

In page 4, line 80; does “multiple birth” associated with childhood anaemia? Or with maternal anaemia? Revise and update.

In page 4, line 84; does “occupation” associated with childhood anaemia? Or with anaemia in adulthood? Revise and update.

In page 4, line 84; “education”, “household wealth status”, and “residence” are just repetition previous statement. Consider deleting them.

In page 4, lines 87-90; in order for the justification to be valid, the author is advised to give the link between reducing child mortality and anaemia prevention and control.

In page 4, lines 90-92; the author justified for this study the cause of “very few studied conducted” on anaemia; while he is using data from 32 studies in the region from the DHS only. This fact is contradicting the author statement. Furthermore, and since the author did not conduct an original research to fill the mentioned gab in knowledge, but only analyzing what have already done, then the limited number of publications could not be considered as an adequate justification for this research. Consider revising the gap in knowledge and restatement of the study justification.

In page 4, lines 92-93; “though the effects of anaemia differ depending on the severity level of anaemia”. You should explain how does anaemia differ depending on severity.

Methods:

In page 5, line 103; what is the cutoff time point for recent and why you chose this cutoff point? DHS; spell it out first then use abbreviation. Were there any inclusion/exclusion criteria used for selecting countries’ data? Are there any country data that was excluded?

In page 5, line 110; have you used any variable in the analysis from other data sets?

In page 5, line 113; how were households been selected?

In page 5, line 114; how were children selected within households?

In page 5, line 115; table 1 should be mentioned in the results section, not in the method section

In page 6, line 123; how was the adjustment to altitude performed? were all studies perform altitude adjustment?

In page 6, line 126; before describing how you assigned variables into the tow levels, explain and justify how and why you came up and restricted your study to the selected variables

In page 6, lines 128-129; how was maternal anaemia measured and defined? What was the haemoglobin level cut point used to define anaemia? Was pregnancy status considered to define anaemia status in women (using different cut points for pregnant and non-pregnant women to define anaemia)? How was the distance to health facility assessed? How was the size of the child being assessed and categories defined?

In page 6, lines 130-132; authors are to mention what were z-scores measuring (e.g. stunting, …) and how they were categorized (what were the cut off points used to define each category)

In page 6, lines 127-133; there are discrepancies between variables listed in the methods section and those listed in tables 2, 3, and 4. Many are in the tables and not in the methods section. Some of which are in some tables but not in other tables. It is not clear what the variables of the study are and how the selection for the model was done.

In page 6, line 138; what other software the author used for the rest of the analysis?

In page 7, lines 154-158; “Which allows the relationship between the explanatory …” This sentence may be better re-phrased to have a more expressive message.

In page 8, line 181; the bi-variable analysis (and related p-values) was not mentioned in the methods section nor presented in the results/tables of results.

In page 9, line 200; “stunted”, “underweight” and “wasted”. Describe the study definitions of these terms in the method section (see above point [In page 6, lines 130-132])

In page 9, lines 203 & 205; “… anaemia among children aged 6 – 59 months in Sub-Saharan Africa was …”. In the results section, use your study population, then at the discussion apply this to the general population if applicable. Consider replacing sub-Saharan Africa with your study population.

In page 9, lines 207-208; the author needs to define what is mother mild, moderate and severe anaemia in the methods section.

In pages 9 & 10, lines 207-212; The author is describing variation between groups. Were these variations statistically significant? In case of using comparisons, you need to show the probability of variability (p-value). Otherwise just describe the data.

In page 10, line 223; “… MOR was 2.03 …” This is different from what presented in Table 4.

In page 10, lines 224-225; “Deviance was used to …” It is better to rephrase this sentence for better understanding.

In page 11, line 241; “… mother education level at no education, primary education …”. Consider using the expression “no formal education” instead of “no education”.

In page 12, lines 287-288; why comparing Sub-Saharan Africa with Brazil and Ecuador? Consider comparison with other continents, regions or previous estimation for the same population rather than countries.

In page 14, lines 309-310; this study findings showed no association between anaemia and birth weight. How could you fix this explanation with your findings?

In page 14, lines 331-332; Transplacental and breast feeding are not common routes of Malaria, Visceral Leishmaniasis and TB transmission!! The evidence used needs to be checked.

Conclusion:

-

Tables:

In Table 1: it is better to group the countries by their sub-regions

In Table 2: many of the variables mentioned in Table 2 were not stated in the methods section, nor used in the tables 3 & 4.

In Table 2: describe abbreviations used in the Table as end note.

In Table 2: use “no formal education” instead of “no education”

In Table 3: the table title “The prevalence and severity of anemia based on the selected child and maternal characteristics in Sub-Saharan Africa”. Consider deleting “selected”, adding “community characteristics”

In Table 3: it is better to keep the order of variable in a way that makes sense. At least keep order the same way in each table.

In Table 3: mention the total number of children per each category of variable

In Table 3: use “no formal education” instead of “no education”

In Table 4: give the p-value for each AOR

In Table 4: “size of the child at birth”. the reference group is better to be the small birth weight (biologically, this makes sense)

General comments:

In the discussion, you need to consider what is the meaning and significance of your results more than justifying your findings using others results. However, the latter is accepted.

Revise grammar, punctuations and sentences clarity.

Reviewer #4: In this article the authors present data on determinants of severity levels of anaemia among children less than 6-59 months in Sub-Saharan Africa using multilevel ordinal logistic regression analysis. The overall prevalence of anaemia was 64.11% of which 26% were found to be mildly anaemic and 34.93 were moderately anaemic with 3% being severely anaemic. The authors also identified a number of factors with increased odd of higher levels of anaemia. These findings are interesting and highlight the relevance of anaemia as a major public health issue in Sub-Sahara Africa not leaving out some of the suggested measures they highlighted which when implemented could help reduce the burden of anaemia. I therefore consider these findings to be of interest in this field and hence worth publishing.

A few Minor points I will like the authors to consider

- The use of the word "Anaemia" and "Anemia" is not correct. They have to decide to use one and not to mix them up. This runs throughout the whole manuscript and need to be corrected.

- Page 4 Line 78: The statement "The finding of previous literature...." is not clear., the authors should consider rephrasing it , 'from' may be better than 'of'

- Page 4 line 90-93. The data used in this analysis were obtained from studies conducted in these regions and therefore to state that state that very few studies have been conducted in this region is not too clear - elaborate a little on this with some supporting references.

- Page 4 line 93. Why do you consider non-anaemic as level of anaemia? Shouldn't these be mild, moderate and severe?

-Page 9 line 195: ...."children got birth", does not read well should consider re-writing the statement like wise what is 'small size at birth' Is it the birth weigh or the actual size of the baby? clarification needed.

- Page 10 Line 217 replace 'cut of' with "cutoff"

- Page 11 the continuous use of the word "higher levels of anaemia" throughout the text is confusing it will be good if the authors use the terms used in their definition-mild, moderate or severe or leave it as the higher odds or lower odds of developing anaemia. Consider revising them.

- Page 14 line 312-Consider revising the statement "...due to teenage mothers are less prepared' likewise line 329 the statement "this is due to the mother is a primary source..." It will have read better if it has been " this is due to the mother being the primary source...."

It will be nice if the discussion section will be relooked at briefly to address some of the points raised .

References

The authors did not meet the referencing style of this journal and therefore I suugest they reformat them to meet the referencing style of this journal

6. PLOS authors have the option to publish the peer review history of their article (what does this mean?). If published, this will include your full peer review and any attached files.

Reviewer #1: No

Reviewer #2: **Yes: **Chigozie Louisa J. Ugwu

Reviewer #3: **Yes: **Khalid Elmardi

Reviewer #4: No

---

## [Author Response · Author response to Decision Letter 0]

8 Feb 2021

PLOS ONE 

Point by point response for editors/reviewers comments 

Manuscript title: Prevalence and determinants of severity levels of anemia among children aged 6-59 months in sub-Saharan Africa: a multilevel ordinal logistic regression analysis

Manuscript ID: PONE-D-20-37075

Dear editor/reviewer. 

Dear all,

We would like to thank you for these constructive, building, and improvable comments on this manuscript that would improve the substance and content of the manuscript. We considered each comment and reviewers on the manuscript thoroughly. Our point-by-point responses for each comment and question are described in detail on the following pages.

Response to reviewers comments

Reviewer#1

1. The manuscript requires intense editing for spelling and grammar.

Authors’ response: Thank you reviewer for the comments. We extensively modified the typographical and grammatical errors with the help of language experts. (See the revised manuscript)

INTRODUCTION

2. This is deficient in the previously reported prevalence rates of childhood anaemia in different African populations. This is needed to establish the burden of anaemia in African children. The burden of anaemia in childhood should be filtered from childhood mortality studies in Africa. All these are required to justify the present study.

Authors’ response: Thank you for the comments. We incorporated the prevalence of anemia in African countries and the childhood mortality related with anemia. (See the revised manuscript, line 67-72 and 90 – 92, page 3-4)

3. Lines 62-63: This is not totally relevant to childhood.

Authors’ response: Thank you for the comments. We removed these sentences in the revised manuscript

4. Line 86: What proportion of childhood mortality is related to anaemia?, Line 90: What “problem” and what “region”?, Lines 95-96: What is the gap in knowledge the study proposed to fill?

Authors’ response: Thank you reviewer for the comments. We considered these comments and modified the manuscript. (See Introduction section, page 4)

METHOD

5. Secondary data analysis. Detailed methodology.

Line 109: What does (46) represent? If it is a reference, it is inappropriate to jump from reference #32 to reference #46. Even the item #46 on the list of references does not tally with the context within the text.

Authors’ response: Thank you for the comments. We wrote the method section in detail and for further we included the link of the data sources as we used secondary data analysis. We updated the references. (See the revised manuscript, page 5-6, line 106 – 127)

6. Line 198: “About” 21666? Figures should be exact.

Lines 132 and 200: What defined which nutritional status?

Lines: 204-206: Prevalence rates should be to a single decimal place.

Line 217: “cut of” or “cut-off”

Authors’ response: Thank you for the comments. We updated the manuscript considering these comments. (See method section, line 157 – 163, page 7 and line 232-240 and line 245, page 10-11 )

Discussion 

7. Lines 295/298: Not specific about which factor increased or decreased with the possible risk of anaemia

Authors’ response: Thank you for the comments. We write separately factors associated with increased or decreased risk of anemia. (See the Discussion section, line 319-325, page 14)

8. Line 313: Inexperience about good infant feeding practices and prevention of infections and infestations may also predispose to deficiencies of haematopoiesis-related nutrients.

Authors’ response: Thank you for the comment. We rewrote it. (See the Discussion section, line 338 – 340, page 15)

9. Line 339: Repetition of “vitamin A”

Lines 342-344: Diarrhea in 6-23 month age group will not explain the higher odd of higher level of anaemia in the 24-59 month age group.

Authors’ response: Thank you for the comments. We removed the repetition of vitamin A, and explain how diarrhea can cause anemia. (See the Discussion section, line 373-377, page 16)

10. Lines 345-347: The role of diarrhoea in childhood anaemia could only be justified if it is frequently recurrent or protracted to cause nutrient losses and nutrient store depletion.

Authors’ response: Thank you for the comment. We accept your statement, as you know diarrhea causes nutrient store depletion when it is recurrent or protracted. In addition, diarrhea is the manifestation of intestinal diseases such as amoebiasis, giardiasis, hookworm, ascariasis, etc, as these diseases are well-known causes of undernutrition in children. So, we are not only talking about the impact of diarrhea but also the diseases that cause diarrhea also responsible for anemia.

11. Lines 349-351: Give examples of intestinal parasitosis known to cause fever, diarrhoea and anaemia in children.

Authors’ response: Thank you for the comment. The commonest intestinal parasitosis that can cause fever, diarrhea, and anemia are visceral leishmaniasis, malaria, hookworm, ascariasis, giardiasis, and amoebiasis. 

12. Lines 352-354: Not clear why wasted children were separated from stunted and underweight children since they are all at risk of higher level of anaemia.

Authors’ response: Thank you for the comment. We write all together in the revised manuscript. (See the Discussion section, line 381-383, page 16)

13. Lines 255-359: Aside deprivation of nutrients required for haematopoiesis, poor nutrition is also associated with poor immunity. Therefore infections and infestations also compound the effects of micronutrient deficiencies.

Authors’ response: Thank you for the comments. We accept this sentence and incorporated in the revised manuscript. (See the Discussion section, line383-386, page 16/17 )

Reviewer#2

1. The paper is well written, the issues tackled are relevant, the methodology applied is appropriate, and the results are properly written. However, the authors were not consistent with using the words anaemia and anemia, sub-Sharan, and Sub-Saharan. Therefore, the entire manuscript requires thorough proofreading and editing.

Authors’ response: Thank you reviewer for the comments. We extensively edited the overall manuscript. (See the revised manuscript)

2. In lines 51, 66, 67, 68, 69, 71, 75, 78, and 79 of pages 3-4, anemia should be changed to anaemia. Please be consistent and edit the entire paper.

In lines 70, 86, and 92 of page 3, sub-Saharan Africa should be written as sub-Saharan Africa. Please check the entire paper.

Authors’ response: Thank you for the comments. We consistently write as anemia and sub-Saharan Africa in the revised manuscript. (See the revised manuscript)

3. Methods

In line 103 of page 4, the authors should refer the readers to Table 1 that shows the list of countries involved in the study. In lines 104 and 106, change Sub to sub. There was no elaboration on the study population and sample size. The inclusion and exclusion criteria were also missing. Highlight information got from the DHS. The paper should also explain how the datasets from the surveys were merged.

Authors’ response: Thank you for the comments. As we stated, we used the DHS data for this study. The study population for this study was children aged 6-59 months and we include under-five children answered the variable anemia status. So, we drop those children who were missing this variable and for further information, we included the DHS website as the data is publically available. We were not merging the data set rather we append the datasets of 32 SSA countries data after extracting similar variables. Because we aim to add observation with similar variables. When we say merging it is all about adding variables whereas appending is adding observation. So, for this study, we have done appending. 

4. The authors failed to list the percentage distribution of anaemia in each of the 32 countries.

Authors’ response: Thank you for the comments. We included the prevalence of anemia across countries. (See Table 1)

5. In line 23 of page 2, the authors wrote anaemia and later changed it to anemia in line 26 of page 2. Hence, they continued interchanging the two words. Thorough editing is highly required.

Authors’ response: Thank you for the concerns. We accept and consistently write as anemia in the revised manuscript. (See the revised manuscript)

6. The -2LLN acronym appeared first in line 179 of page 8; therefore, the authors should write the full meaning at the location in the article.

Authors’ response; Thank you for the comments. We write in full term. (See the revised manuscript, line 205, page 9)

7. On page 7, the authors should clearly explain the analytical strategy and process, providing references.

Authors’ response: Thank you for the comment. We stated in the method section in detail. For the descriptive results, we reported using numerical figures and percentages. For associated factors, we applied multilevel ordinal logistic regression models. (See the Method section)

8. Results and discussion

In line 194 of page 9, the authors mention the percentage distribution of anaemia among children in East Africa. It is also necessary that the percentage distribution of childhood anaemia in West Africa, Central Africa, and Southern Africa should be mentioned.

Authors’ response: Thank you for the comments. We reported the prevalence of anemia in all the SSA regions. (See the Result section, line 220-221, page 10)

9. In line 202 of page 10, change Sub to sub.

In lines 217, 219, 221, 222, and 227 of page 10, change anemia to anaemia. Please be consistent and edit the whole manuscript.

In line 272 of page 12, South Africa should be changed to Southern Africa.

Correct the anemia in line 279 and Sub-Sharan Africa in line 280 of page 12.

Lines 287, 290, 291, 296, and 307 of page 13 should also be corrected.

Correct the anemia in lines 326 and 327 of page 14.

Lines 334 and 335 of page 15 should also be corrected.

Correct the anemia in lines 372 and 381 of page 16.

Authors’ response: Thank you for the comments. We made it consistent in the revised manuscript as anemia and sub-Saharan Africa. (See the revised manuscript)

10. A table of unadjusted ORs is missing.

Authors’ response: Thank you for the comments. We have fitted a multilevel ordinal logistic regression model and we fit four models. As you can see in Table 4 we presented the AOR for four models, and if we write the Crude Odds Ratio (COR) the table will be bulky and hard to catch. If we were fitted the ordinary logistic regression we are expected to report the COR but we have fitted the multilevel logistic regression model, and if we report the COR, it will be tedious. If this doesn't convince you we are ready to report it.

Reviewer#3

Abstract 

1. Background:

In page 2, lines 26-27; the sentence “Though anemia is preventable, it remains ...” looks odd. It is distracting from the main scope of the study. Is better to be removed.

There is no valid justification stated for this research. Thus, the word “therefore” in the sentence “Therefore, this study ...” (in Page 2, line 27) seems strange. It is advised either to give reasonable justification or to remove it.

Authors’ response: Thank you for the comments. We rewrote it. (See the revised manuscript)

2. Methods:

In page 2, line 30; “This study was based on the most recent Demographic and Health Survey data ...” what is the limit/definition of recent. How far recent Demographic and Health Survey data are accepted to be included in this study. Please define

Authors’ response: Thank you for the comments. In this study, we excluded the DHS survey of countries Eriteria as the last DHS was conducted in 1995. Therefore, we included countries with DHS survey after 2000 means following the MDGs. Fortunately, the last DHS survey of the included 32 countries was conducted from 2005 to 2016. We say recent means the last DHS of countries after 2000 as many public health programs were implemented following 2000. 

3. In page 2, lines 32-33; “... and the ordinal nature of anaemia”. What is this ordinal nature? “... a multilevel ordinal logistic regression model was applied.” What is the study outcome?

In page 2, lines 33-34; “Proportional odds assumption was tested by Brant test ...” what is the purpose of using this test?

Authors’response: Thank you for the comments. As we know the scale of measurement of the outcome variable that was level of anemia was categorized as non-anemic, mild, moderate, and severe anemia. Therefore, the level of anemia is an ordinal variable as it has a kind of order. So, as you know the choice of the method of analysis is depending on the outcome variable (level of anemia), the ordinal logistic regression model is the appropriate method of analysis since the response variable has more than two choices. Besides, as our data source was the DHS data, the data has hierarchical nature and so, study subjects within the same cluster might share similar characteristics to individuals from another cluster. Therefore, we applied an advanced statistical model to take into account the clustering effect and apply a multilevel ordinal logistic regression model. In the ordinal logistic regression model, there is one basic assumption that is proportional odds assumption/parallel line assumptions. The proportional odds assumption is used to check whether the ordinal logistic regression model is the best-fitted or not. If this assumption is violated, indicates that the proportional odds model is not the appropriate model for the data and therefore, we have to consider the partial odds model or multinomial logistic regression model. In this study, the proportional odds assumption was satisfied (p-value>0.05) and indicates the multilevel ordinal logistic model is the well-fitted model for the data. 

4. In page 2, lines 36; “... since the models were nested models.” This part of the sentence is better be deleted. Independent variables were not described. You don’t need to list them, but only describe them in summary; therefore, the reader could have an idea about what you are looking at.

Authors’ response: Thank you for the comments. We removed the sentences and rewrite them. (See the revised manuscript)

5. Results:

In page 2, lines 40-41; “... prevalence of anaemia among children aged 6-59 months in sub-Saharan Africa...” in the results, use your study population and do not apply your results to the general population. So, replace sub-Saharan Africa by another expression/word to reflect your own findings.

Authors’ response: Thank you for the comments. We accept the comments and rewrote it.

6. In page 2, lines 45; “... were significantly associated with an increased odd of higher...” correct the grammar of “an increased odd” to “increased odds”

Authors’ response: Thank you for the comments. We corrected it.

7. While the authors used an ordinal logistic regression, nothing is mentioned about the order of severity in the results?!!! 

Conclusion:

Nothing is mentioned about the level of severity in the conclusion.

Authors’ response: Thank you for your concern. As you can see in the result section we presented the prevalence of anemia in the order of severity whereas, in the ordinal logistic regression we reported a single OR since the proportional assumption was satisfied and the OR was constant across categories. If the proportional odds assumption was violated we were reporting the separate odds ratio for each order. Now, we mentioned the levels of anemia in the conclusion section.

8. The main article:

Background:

In page 3, lines 49-50; “... World Health 60 Organization (WHO) defines childhood anemia as a Hb concentration below 110 g/L.” The WHO has three cut-off levels for defining anaemia in childhood depending on age group. Please revise and correct your statement. Ref. [3] is not a WHO publication. If you are referring your statement to the WHO as a reference organization for setting standard definitions, then you should cite reference from that organization itself and not what others stated what WHO had said! Please revise.

Authors’ response: Thank you for the comments. We corrected the references. See the revised manuscript.

9. In page 3, lines 62-66; the author stated many causes that he did not studied. Stating such issues in the background without being related to the study seems redundancy and could distract the reader from the target of the study. Consider deleting it.

Authors’ response: Thank you for the comments. We removed these sentences in the revised manuscript. (See revised manuscript)

10. In page 3, lines 64-65; the word “hemoglobinopathies” is duplicated

In page 4, lines 79-84; the author mentioned - from previous studies - many factors associated with anaemia; however, some of them (infectious diseases (malaria, hookworm)) were not included in this study! Explain why.

Authors’ response: Thank you for the comments. We removed the duplicated word. The data source for this study was DHS data and in this dataset, the variables such as infectious diseases and other clinical factors were not found. That is why we did not incorporate as a variable in this study. Besides, we acknowledge the limitation section of the study.

11. In page 4, line 80; does “multiple birth” associated with childhood anaemia? Or with maternal anaemia? Revise and update. In page 4, line 84; does “occupation” associated with childhood anaemia? Or with anaemia in adulthood? Revise and update.

Authors’ response: Thank you for the concerns. As you know multiple births such as twin births are more likely prone to malnutrition, anemia, and other poor outcomes, they are more likely to be anemic than singletons. Regarding occupation, as you know maternal occupation is closely linked with the wealth status of the household and therefore, the woman who had occupation are more likely to afford the costs for child care and nutrition. We included these variables as previous literature reported as significantly associated factors with childhood anemia.

12. In page 4, line 84; “education”, “household wealth status”, and “residence” are just repetition previous statement. Consider deleting them. In page 4, lines 87-90; in order for the justification to be valid, the author is advised to give the link between reducing child mortality and anaemia prevention and control.

Authors’ response: Thank you for the comments. We removed the repetition. Besides, we incorporated how anemia prevention and control reduces child mortality. Anemia is responsible for the death of million of children and therefore, working on anemia can save millions of children. (See the revised manuscript)

13. In page 4, lines 90-92; the author justified for this study the cause of “very few studied conducted” on anaemia; while he is using data from 32 studies in the region from the DHS only. This fact is contradicting the author statement. Furthermore, and since the author did not conduct an original research to fill the mentioned gab in knowledge, but only analyzing what have already done, then the limited number of publications could not be considered as an adequate justification for this research. Consider revising the gap in knowledge and restatement of the study justification. In page 4, lines 92-93; “though the effects of anaemia differ depending on the severity level of anaemia”. You should explain how does anaemia differ depending on severity.

Authors’ response: Thank you for the comments. As we stated in the background section, we justify the significance of the study from public health perspective and methodological perspective. Regarding the public health perspective, this study was based on the pooled DHS data of 32 sub-Saharan African countries with a very large sample size and this could increase the power of the study and the estimate can be generalized. Besides, the use of multilevel approach, is mainly concentrated on the ecological approach of epidemiology as it can take into account the neighborhood effect, and the result can give the overall picture of SSA. Regarding the methodological perspective, as you can see previously published literatures treat anemia as a binary outcome by categorizing no/yes but as you can understand treating mild, moderate, and severe anemia as yes is not statistically appropriate since there is the loss of information because the factor responsible for mild anemia may not be similar with the factor that can cause severe anemia. Therefore, we applied the multilevel ordinal logistic regression model to get a reliable estimate and avoid loss of information. (See the Background section)

14. Methods:

In page 5, line 103; what is the cutoff time point for recent and why you chose this cutoff point? DHS; spell it out first then use abbreviation. Were there any inclusion/exclusion criteria used for selecting countries’ data? Are there any country data that was excluded? In page 5, line 110; have you used any variable in the analysis from other data sets? In page 5, line 113; how were households been selected? In page 5, line 114; how were children selected within households?

Authors’ response: Thank you for the comments. We spell out DHS and regarding the cutoff point just used the countries with the last DHS survey after 2000 by relating it with MDG and in our study, we included survey’s from 2005-2016 and we considered them as a factor and were not significant. We reported the DHS databases to link for further methodological procedures. In the selected households the most recent children were selected for this study. (See the Method section)

15. In page 5, line 115; table 1 should be mentioned in the results section, not in the method section

Authors’ response: Thank you. We mentioned it in the result section. (See the Result section)

16. In page 6, line 123; how was the adjustment to altitude performed? were all studies perform altitude adjustment?

Authors’ response: Thank you for the comments. We included it in the method section. (See the method section, line 135-143, page 6)

17. In page 6, line 126; before describing how you assigned variables into the tow levels, explain and justify how and why you came up and restricted your study to the selected variables

Authors’ response: Thank you for the comments. As we stated above, we conducted secondary data analysis using DHS as a data source, and we considered these variables for analysis as these are the variables that we can access from DHS, and we recode based on literature.

18. In page 6, lines 128-129; how was maternal anaemia measured and defined? What was the haemoglobin level cut point used to define anaemia? Was pregnancy status considered to define anaemia status in women (using different cut points for pregnant and non-pregnant women to define anaemia)? How was the distance to health facility assessed? How was the size of the child being assessed and categories defined?

Authors’ response: Thank you for the comments. The cutoff point for maternal anemia was used differently for pregnant and non-pregnant mothers as we can see in the DHS guideline. Regarding health distance facility was assessed subjectively asking a question how do you see the distance to reach health facility and they responded as a big problem and not a big problem. Whereas, about the size of the child at birth, it was assessed by asking mothers what was the size of the child at birth and they responded as very small, small, average, large, and very large.

19. In page 6, lines 130-132; authors are to mention what were z-scores measuring (e.g. stunting, …) and how they were categorized (what were the cut off points used to define each category).

Authors’ response: Thank you for the comments. We included it in the revised manuscript. (See Method section, line 161-163, page 7)

20. In page 6, lines 127-133; there are discrepancies between variables listed in the methods section and those listed in tables 2, 3, and 4. Many are in the tables and not in the methods section. Some of which are in some tables but not in other tables. It is not clear what the variables of the study are and how the selection for the model was done.

Authors’ response: Thank you for the comments. We resolve the discrepancies between variables in the method and result section. For the multilevel ordinal logistic regression analysis, first, we select variables which have p-values less than 0.2 in the bi-variable multilevel ordinal logistic regression analysis. Then, we used these variables for the multivariable multilevel ordinal regression analysis. Then, we have checked the proportional odds assumption to choose which ordinal model is appropriate, and the proportional odds model was the appropriate model for the data (p-value>0.05). Then, we checked whether there is a clustering effect as the DHS data has hierarchical nature. Finally, we built four models(null model, model I, model II, and model III) and we compared using deviance.

21. In page 6, line 138; what other software the author used for the rest of the analysis?

In page 7, lines 154-158; “Which allows the relationship between the explanatory …” This sentence may be better re-phrased to have a more expressive message.

Authors’ response: Thank you for the comments. We used STATA version 14 statistical software for the overall analysis. We write it. (See the revised manuscript, line 164 – 210, page 7-9)

22. In page 8, line 181; the bi-variable analysis (and related p-values) was not mentioned in the methods section nor presented in the results/tables of results.

In page 9, line 200; “stunted”, “underweight” and “wasted”. Describe the study definitions of these terms in the method section (see above point [In page 6, lines 130-132]).

Authors’ response: Thank you for the comments. As we mentioned in the method section, we conduct the bi-variable multilevel ordinal logistic regression analysis to select variables for the multivariable multilevel ordinal logistic regression analysis, and variables with a p-value less than 0.2 were considered for the multivariable analysis. In addition, we defined the variables stunted, underweight, and wasted. (See the method section, line 160-162, page 7)

23. In page 9, lines 203 & 205; “… anaemia among children aged 6 – 59 months in Sub-Saharan Africa was …”. In the results section, use your study population, then at the discussion apply this to the general population if applicable. Consider replacing sub-Saharan Africa with your study population.

Authors’ response: Thank you for the comments. We accept your comment and revised it. (See the revised manuscript)

24. In page 9, lines 207-208; the author needs to define what is mother mild, moderate and severe anaemia in the methods section

Authors’ response: Thank you for the comments. We incorporated in the revised manuscript. (See the revised manuscript, Method section, line 162 – 165, page 7)

25. In pages 9 & 10, lines 207-212; The author is describing variation between groups. Were these variations statistically significant? In case of using comparisons, you need to show the probability of variability (p-value). Otherwise just describe the data.

Authors’ response: Thank you for the comments. Here we need to show whether there is a difference in the prevalence of severity level of anemia across variable categories and we assessed whether this difference is significant or not using the multilevel ordinal logistic regression. As you mentioned we aim to describe severity levels of anemia across categories, that is why we did not present the p-value.

26. In page 10, line 223; “… MOR was 2.03 …” This is different from what presented in Table 4.

In page 10, lines 224-225; “Deviance was used to …” It is better to rephrase this sentence for better understanding.

Authors’ response: Thank you for the comments. Regarding MOR, it was an editorial error and we corrected it. Concerning Deviance, based on the suggestions we rephrase it. (See the revised manuscript, line 253 – 255, page 11)

27. In page 11, line 241; “… mother education level at no education, primary education …”. Consider using the expression “no formal education” instead of “no education”.

Authors’ response: Thank you for the comments. We modified it. (See revised manuscript)

28. In page 12, lines 287-288; why comparing Sub-Saharan Africa with Brazil and Ecuador? Consider comparison with other continents, regions or previous estimation for the same population rather than countries.

Authors’ response: Thank you for the comments. You are right but we compare our findings with these studies because we didn’t get studies at the sub-Saharan Africa level using DHS data at the SSA level. In addition, we compared this estimate with other countries in Europe and Latin American countries to detect the differences.

29. In page 14, lines 309-310; this study findings showed no association between anaemia and birth weight. How could you fix this explanation with your findings?

Authors’ response: Thank you for the comments. The factor we considered for this study was not birth weight rather birth size was included. Since birth weight was not found for the majority of the children, and data on mothers' perceived birth size was collected. And in our analysis birth size was not significant, probably might be because of the bias introduced by mothers as it was a subjective measurement. 

30. In page 14, lines 331-332; Transplacental and breast feeding are not common routes of Malaria, Visceral Leishmaniasis and TB transmission!! The evidence used needs to be checked.

Authors’ response: Thank you for the comments. Though the incidence is too low malaria, TB, HIV/AIDS and Vl can have the potential to transmit vertically from mothers to fetus especially when the placenta is infected. As you know congenital malaria and congenital tuberculosis are reported in literature. Now, we mentioned HIV/AIDS and malaria. (See the revised manuscript)

31. Conclusion

Tables:

In Table 1: it is better to group the countries by their sub-regions

In Table 2: many of the variables mentioned in Table 2 were not stated in the methods section, nor used in the tables 3 & 4.

In Table 2: describe abbreviations used in the Table as end note.

In Table 2: use “no formal education” instead of “no education”

Authors’ response: Thank you for the comments. We corrected it. (See Table 1 and 2)

32. In Table 3: the table title “The prevalence and severity of anemia based on the selected child and maternal characteristics in Sub-Saharan Africa”. Consider deleting “selected”, adding “community characteristics”

In Table 3: it is better to keep the order of variable in a way that makes sense. At least keep order the same way in each table.

In Table 3: mention the total number of children per each category of variable

In Table 3: use “no formal education” instead of “no education”

Authors’ response: Thank you for the comments. We modified the manuscript considering the concerns. (See the revised manuscript)

33. In Table 4: give the p-value for each AOR

In Table 4: “size of the child at birth”. the reference group is better to be the small birth weight (biologically, this makes sense)

Authors’ response: Thank you for the comments. We did not report p-value for each AOR because OR is more informative than p-value as it can give information about the sample size in addition to the significance and strength of association. Besides, we also represent the p-value using asterisk, as you can see, we presented in the footnotes of Table 4.

34. General comments:

In the discussion, you need to consider what is the meaning and significance of your results more than justifying your findings using others results. However, the latter is accepted. Revise grammar, punctuations and sentences clarity.

Authors’ response: Thank you for the comments. We extensively edited for any typographical and grammatical errors. (See the revised manuscript)

Reviewer # 4

1. In this article the authors present data on determinants of severity levels of anaemia among children less than 6-59 months in Sub-Saharan Africa using multilevel ordinal logistic regression analysis. The overall prevalence of anaemia was 64.11% of which 26% were found to be mildly anaemic and 34.93 were moderately anaemic with 3% being severely anaemic. The authors also identified a number of factors with increased odd of higher levels of anaemia. These findings are interesting and highlight the relevance of anaemia as a major public health issue in Sub-Sahara Africa not leaving out some of the suggested measures they highlighted which when implemented could help reduce the burden of anaemia. I therefore consider these findings to be of interest in this field and hence worth publishing.

Authors’ response: Thank you for the comments. We suggested the measures should be taken to reduce the prevalence of childhood anemia and its consequence in the discussion section of the manuscript. 

2. A few Minor points I will like the authors to consider

The use of the word "Anaemia" and "Anemia" is not correct. They have to decide to use one and not to mix them up. This runs throughout the whole manuscript and need to be corrected.

 Page 4 Line 78: The statement "The finding of previous literature...." is not clear., the authors should consider rephrasing it , 'from' may be better than 'of'

Authors’ response: Thank you for the comments. We modified and write consistently. (See the revised manuscript)

3. - Page 4 line 90-93. The data used in this analysis were obtained from studies conducted in these regions and therefore to state that state that very few studies have been conducted in this region is not too clear - elaborate a little on this with some supporting references.

Authors’ response: Thank you for the comments. We elaborate it. (See the Background section, line 95-105, page 4/5 )

4. Page 4 line 93. Why do you consider non-anaemic as level of anaemia? Shouldn't these be mild, moderate and severe?

-Page 9 line 195: ...."children got birth", does not read well should consider re-writing the statement like wise what is 'small size at birth' Is it the birth weigh or the actual size of the baby? clarification needed.

Authors’ response: Thank you for the comments. As we know the scale of measurement of the outcome variable that was level of anemia was categorized as non-anemic, mild, moderate and severe anemia. Therefore, the level of anemia is ordinal variable as it has a kind of ordering. So, as you know the choice of the method of analysis is depending on the outcome variable (level of anemia), the ordinal logistic regression model is the appropriate method of analysis since the response variable has more than two choices. Whereas, about size of child at birth, it was assessed by asking mothers what was the size of the child at birth and they responded as very small, small, average, large, and very large. So, birth size is the about mothers perceived birth size and we recode it as small (very small plus small), average and large (large plus very large).

5. - Page 10 Line 217 replace 'cut of' with "cutoff"

- Page 11 the continuous use of the word "higher levels of anaemia" throughout the text is confusing it will be good if the authors use the terms used in their definition-mild, moderate or severe or leave it as the higher odds or lower odds of developing anaemia. Consider revising them.

Authors’ response: Thank you for the comments. We replace it and we modified it. 

6. - Page 14 line 312-Consider revising the statement "...due to teenage mothers are less prepared' likewise line 329 the statement "this is due to the mother is a primary source..." It will have read better if it has been " this is due to the mother being the primary source...."

Authors’ response: Thank you for the comments. We revised it.

7. References

The authors did not meet the referencing style of this journal and therefore I suugest they reformat them to meet the referencing style of this journal

Authors’ response: Thank you for the comments. We modified it.

---

## [Decision Letter · Decision Letter 1]

8 Mar 2021

PONE-D-20-37075R1

Prevalence and determinants of severity levels of anemia among children aged 6-59 months in sub-Saharan Africa: a multilevel ordinal logistic regression analysis

PLOS ONE

Dear Dr. Tesema,

Thank you for submitting your manuscript to PLOS ONE. After careful consideration, we feel that it has merit but does not fully meet PLOS ONE’s publication criteria as it currently stands. Therefore, we invite you to submit a revised version of the manuscript that addresses the points raised during the review process.

We look forward to receiving your revised manuscript.

Kind regards,

Frank T. Spradley

Academic Editor

PLOS ONE

Journal Requirements:

Reviewers' comments:

Reviewer's Responses to Questions

**Comments to the Author**

1. If the authors have adequately addressed your comments raised in a previous round of review and you feel that this manuscript is now acceptable for publication, you may indicate that here to bypass the “Comments to the Author” section, enter your conflict of interest statement in the “Confidential to Editor” section, and submit your "Accept" recommendation.

Reviewer #2: All comments have been addressed

Reviewer #3: (No Response)

Reviewer #4: All comments have been addressed

2. Is the manuscript technically sound, and do the data support the conclusions?

Reviewer #2: (No Response)

Reviewer #3: Yes

Reviewer #4: Yes

3. Has the statistical analysis been performed appropriately and rigorously? 

Reviewer #2: (No Response)

Reviewer #3: Yes

Reviewer #4: Yes

4. Have the authors made all data underlying the findings in their manuscript fully available?

Reviewer #2: (No Response)

Reviewer #3: Yes

Reviewer #4: Yes

5. Is the manuscript presented in an intelligible fashion and written in standard English?

Reviewer #2: (No Response)

Reviewer #3: Yes

Reviewer #4: Yes

6. Review Comments to the Author

Reviewer #2: (No Response)

Reviewer #3: Previous reviewer comment

Point 8. The main article:

In background section:

Reviewer comment: In page 3, lines 49-50; “... World Health Organization (WHO) defines childhood anemia as a Hb concentration below 110 g/L.” The WHO has three cut-off levels for defining anaemia in childhood depending on age group. Please revise and correct your statement. Please revise.

Authors’ response: Thank you for the comments. We corrected the references. See the revised manuscript

Reviewer feedback 1: childhood do not denote for under five years old children. As mentioned above this cut-off level of haemoglobin is not valid for all childhood age categories. It is only valid for children under the age of 5 years. Author has corrected the reference but not the sentence. Kindly, correct your statement about this. Moreover, this reference used from the WHO is less relevant. It is about iron deficiency anaemia. The WHO has other resources which tell exactly what is the different cut-off levels for defining anaemia in different age, population, and physiological status. Furthermore, the reference used was not properly cited.

Previous reviewer comment

Point 13. In page 4, lines 90-92; the author justified for this study the cause of “very few studied conducted” on anaemia; while he is using data from 32 studies in the region from the DHS only. This fact is contradicting the author statement. Furthermore, and since the author did not conduct an original research to fill the mentioned gab in knowledge, but only analyzing what have already done, then the limited number of publications could not be considered as an adequate justification for this research. Consider revising the gap in knowledge and restatement of the study justification.

Authors’ response: Thank you for the comments. As we stated in the background section, we justify the significance of the study from public health perspective and methodological perspective. Regarding the public health perspective, this study was based on the pooled DHS data of 32 sub-Saharan African countries with a very large sample size and this could increase the power of the study and the estimate can be generalized. Besides, the use of multilevel approach, is mainly concentrated on the ecological approach of epidemiology as it can take into account the neighborhood effect, and the result can give the overall picture of SSA. Regarding the methodological perspective, as you can see previously published literatures treat anemia as a binary outcome by categorizing no/yes but as you can understand treating mild, moderate, and severe anemia as yes is not statistically appropriate since there is the loss of information because the factor responsible for mild anemia may not be similar with the factor that can cause severe anemia. Therefore, we applied the multilevel ordinal logistic regression model to get a reliable estimate and avoid loss of information. (See the Background section).

Reviewer feedback 2: these justifications are acceptable. But the statement of “very few studied conducted” is not acceptable. This undermines others’ work. In fact, there are many studies on anaemia prevalence and its determinants available as articles and other resource literature. You can include what you have stated above in your article, but you need to revises the phrase about availability of “very few studies on anaemia” as a justification.

Previous reviewer comment

Point 14. Methods:

In page 5, line 113; how were households been selected? In page 5, line 114; how were children selected within households?

Authors’ response: Thank you for the comments. …. We reported the DHS databases to link for further methodological procedures. In the selected households the most recent children were selected for this study. (See the Method section)

Reviewer feedback 3: In your study, you need to describe the methodology clearly and in details. Sample selection method should receive special concern and detailed description to ensure that the study design is properly fit the data and its collection method. Yes, you could refer readers to the website/other reference for more details, but the details that describe your study. In your case, describing selection of the sample at each level has its implication on choosing the multilevel model for your analysis and the way that you considered the levels. More over describing the methodology will give an indication about the extent to which methods are homogenous in all countries and that they could be used in your study without the need for prior manipulation/re-arrangement. If children were not randomly selected then how can you explain avoiding bias in selection.

So, you need to describe how households were selected. The same apply for child selection with each household.

Previous reviewer comment

Point 18. In page 6, lines 128-129; … How was the distance to health facility assessed? How was the size of the child being assessed and categories defined?

Authors’ response: Thank you for the comments. …. Regarding health distance facility was assessed subjectively asking a question how do you see the distance to reach health facility and they responded as a big problem and not a big problem. Whereas, about the size of the child at birth, it was assessed by asking mothers what was the size of the child at birth and they responded as very small, small, average, large, and very large.

Reviewer feedback 4: The author needs to mention these explanations the article to clarify these issues. However, in response to point 29 in the discussion section, this subjective measurement of size of the baby creates incomparable results with what is available in the literature as stated by the author! So, the author may think of removing variables with non-specific definitions/measurement.

Related to this is that, “distance to health facility” was removed from the text while it was still there in table 4. However, it is not mentioned in tables 2 and 3! Kindly, assure consistency and cohesion.

More issues related to this include mismatch between what is in the text as variables and what is in table 2 and 3. PNC visit was mentioned in table 2 where it is not there in the article. For table 3, many important variables disappeared from table 3 while they were mentioned in table 2. These includes source of drinking water, sex of the head of the household, wanted pregnancy, employment status of the mother, smoking status of the mother, cough, receiving antiparasitic drugs and vit. A supplementation.

Reviewer #4: (No Response)

7. PLOS authors have the option to publish the peer review history of their article (what does this mean?). If published, this will include your full peer review and any attached files.

Reviewer #2: **Yes: **Chigozie Louisa J. Ugwu

Reviewer #3: **Yes: **Khalid Elmardi

Reviewer #4: No

---

## [Author Response · Author response to Decision Letter 1]

13 Mar 2021

Point by point response for editors/reviewers comments 

PLOS ONE Journal 

Manuscript title: Prevalence and determinants of severity levels of anemia among children aged 6-59 months in sub-Saharan Africa: a multilevel ordinal logistic regression analysis

Manuscript ID: PONE-D-20-37075R1

Dear editor. 

Dear all,

We would like to thank you for these constructive, building, and improvable comments on this manuscript that would improve the substance and content of the manuscript. We considered each comment and clarification questions of editors and reviewers on the manuscript thoroughly. Our point-by-point responses for each comment and question are described in detail on the following pages. Further, the details of changes were shown by track changes in the supplementary document attached.

Response to Reviewer comment

Reviewer#3 

1. - Point 8. The main article:

In background section:

Reviewer comment: In page 3, lines 49-50; “... World Health Organization (WHO) defines childhood anemia as a Hb concentration below 110 g/L.” The WHO has three cut-off levels for defining anaemia in childhood depending on age group. Please revise and correct your statement. Please revise.

Authors’ response: Thank you for the comments. We corrected the references. See the revised manuscript

Reviewer feedback 1: childhood do not denote for under five years old children. As mentioned above this cut-off level of haemoglobin is not valid for all childhood age categories. It is only valid for children under the age of 5 years. Author has corrected the reference but not the sentence. Kindly, correct your statement about this. Moreover, this reference used from the WHO is less relevant. It is about iron deficiency anaemia. The WHO has other resources which tell exactly what is the different cut-off levels for defining anaemia in different age, population, and physiological status. Furthermore, the reference used was not properly cited.

Authors’ response: Thank you for the comment. We accept the comment and rewrite as anemia among under-five children. (See the revised manuscript)

2. Previous reviewer comment

Point 13. In page 4, lines 90-92; the author justified for this study the cause of “very few studied conducted” on anaemia; while he is using data from 32 studies in the region from the DHS only. This fact is contradicting the author statement. Furthermore, and since the author did not conduct an original research to fill the mentioned gab in knowledge, but only analyzing what have already done, then the limited number of publications could not be considered as an adequate justification for this research. Consider revising the gap in knowledge and restatement of the study justification.

Authors’ response: Thank you for the comments. As we stated in the background section, we justify the significance of the study from public health perspective and methodological perspective. Regarding the public health perspective, this study was based on the pooled DHS data of 32 sub-Saharan African countries with a very large sample size and this could increase the power of the study and the estimate can be generalized. Besides, the use of multilevel approach, is mainly concentrated on the ecological approach of epidemiology as it can take into account the neighborhood effect, and the result can give the overall picture of SSA. Regarding the methodological perspective, as you can see previously published literatures treat anemia as a binary outcome by categorizing no/yes but as you can understand treating mild, moderate, and severe anemia as yes is not statistically appropriate since there is the loss of information because the factor responsible for mild anemia may not be similar with the factor that can cause severe anemia. Therefore, we applied the multilevel ordinal logistic regression model to get a reliable estimate and avoid loss of information. (See the Background section).

Reviewer feedback 2: these justifications are acceptable. But the statement of “very few studied conducted” is not acceptable. This undermines others’ work. In fact, there are many studies on anaemia prevalence and its determinants available as articles and other resource literature. You can include what you have stated above in your article, but you need to revises the phrase about availability of “very few studies on anaemia” as a justification.

Authors’ response: Thank you for the comments. We accept the suggestions and modified the revised manuscript. (See the revised manuscript)

3. Point 14. Methods:

In page 5, line 113; how were households been selected? In page 5, line 114; how were children selected within households?

Authors’ response: Thank you for the comments. …. We reported the DHS databases to link for further methodological procedures. In the selected households the most recent children were selected for this study. (See the Method section)

Reviewer feedback 3: In your study, you need to describe the methodology clearly and in details. Sample selection method should receive special concern and detailed description to ensure that the study design is properly fit the data and its collection method. Yes, you could refer readers to the website/other reference for more details, but the details that describe your study. In your case, describing selection of the sample at each level has its implication on choosing the multilevel model for your analysis and the way that you considered the levels. More over describing the methodology will give an indication about the extent to which methods are homogenous in all countries and that they could be used in your study without the need for prior manipulation/re-arrangement. If children were not randomly selected then how can you explain avoiding bias in selection.

So, you need to describe how households were selected. The same apply for child selection with each household.

Authors’ response: Thank you for the comments. We included the statement about how the households selected under the method section of the manuscript. (See the revised manuscript)

4. Point 18. In page 6, lines 128-129; … How was the distance to health facility assessed? How was the size of the child being assessed and categories defined?

Authors’ response: Thank you for the comments. …. Regarding health distance facility was assessed subjectively asking a question how do you see the distance to reach health facility and they responded as a big problem and not a big problem. Whereas, about the size of the child at birth, it was assessed by asking mothers what was the size of the child at birth and they responded as very small, small, average, large, and very large.

Reviewer feedback 4: The author needs to mention these explanations the article to clarify these issues. However, in response to point 29 in the discussion section, this subjective measurement of size of the baby creates incomparable results with what is available in the literature as stated by the author! So, the author may think of removing variables with non-specific definitions/measurement.

Related to this is that, “distance to health facility” was removed from the text while it was still there in table 4. However, it is not mentioned in tables 2 and 3! Kindly, assure consistency and cohesion.

More issues related to this include mismatch between what is in the text as variables and what is in table 2 and 3. PNC visit was mentioned in table 2 where it is not there in the article. For table 3, many important variables disappeared from table 3 while they were mentioned in table 2. These includes source of drinking water, sex of the head of the household, wanted pregnancy, employment status of the mother, smoking status of the mother, cough, receiving antiparasitic drugs and vit. A supplementation.

Authors’ response: Thank you for the comments. regrading distance to health facility, we have removed from the manuscript as it is more of subjective, and we included PNC in the variable of the study section but was not included in the model as it has p-value>0.2 in the bi-variable analysis. Regrading birth size of a child, as you know birth weight of the child was assessed in two ways such as mothers perceived birth size of a child and measured birth weight of the child but the measured weight were missed in more than 80 percent of child’s. So, in developing countries mothers perceived size is commonly used and it has more than 90 percent overall agreement with the measured Birth weight. Besides, we acknowledge in the limitation section.

For Table 3, we presented the severity level of anemia by selecting the commonly reported predictors of anemia that is why we missed some of the variables in Table three.

---

## [Decision Letter · Decision Letter 2]

29 Mar 2021

Prevalence and determinants of severity levels of anemia among children aged 6-59 months in sub-Saharan Africa: a multilevel ordinal logistic regression analysis

PONE-D-20-37075R2

Dear Dr. Tesema,

We’re pleased to inform you that your manuscript has been judged scientifically suitable for publication and will be formally accepted for publication once it meets all outstanding technical requirements.

Kind regards,

Frank T. Spradley

Academic Editor

PLOS ONE

Reviewers' comments:

Reviewer's Responses to Questions

**Comments to the Author**

1. If the authors have adequately addressed your comments raised in a previous round of review and you feel that this manuscript is now acceptable for publication, you may indicate that here to bypass the “Comments to the Author” section, enter your conflict of interest statement in the “Confidential to Editor” section, and submit your "Accept" recommendation.

Reviewer #3: All comments have been addressed

2. Is the manuscript technically sound, and do the data support the conclusions?

Reviewer #3: (No Response)

3. Has the statistical analysis been performed appropriately and rigorously? 

Reviewer #3: (No Response)

4. Have the authors made all data underlying the findings in their manuscript fully available?

Reviewer #3: (No Response)

5. Is the manuscript presented in an intelligible fashion and written in standard English?

Reviewer #3: (No Response)

6. Review Comments to the Author

Reviewer #3: (No Response)

7. PLOS authors have the option to publish the peer review history of their article (what does this mean?). If published, this will include your full peer review and any attached files.

Reviewer #3: **Yes: **Khalid Elmardi

---

## [Editor Report · Acceptance letter]

13 Apr 2021

PONE-D-20-37075R2 

Prevalence and determinants of severity levels of anemia among children aged 6-59 months in sub-Saharan Africa: a multilevel ordinal logistic regression analysis 

Dear Dr. Tesema:

I'm pleased to inform you that your manuscript has been deemed suitable for publication in PLOS ONE. Congratulations! Your manuscript is now with our production department. 

Kind regards, 

on behalf of

Dr. Frank T. Spradley 

Academic Editor

PLOS ONE